# Effective Robustness against Natural Distribution Shifts for Models with Different Training Data

**Zhouxing Shi**[*]
UCLA
zshi@cs.ucla.edu

**Nicholas Carlini**
Google Research
ncarlini@google.com

**Ananth Balashankar**
Google Research
ananthbshankar@google.com

**Ludwig Schmidt**
University of Washington
schmidt@cs.washington.edu

**Cho-Jui Hsieh**
Google, UCLA
chohsieh@cs.ucla.edu

**Alex Beutel**[*]
OpenAI
alexb@openai.com

**Yao Qin**
UCSB, Google Research
yaoqin@ucsb.edu

## Abstract

"Effective robustness" measures the extra out-of-distribution (OOD) robustness beyond what can be predicted from the in-distribution (ID) performance. Existing effective robustness evaluations typically use a single test set such as ImageNet to evaluate the ID accuracy. This becomes problematic when evaluating models trained on different data distributions, e.g., comparing models trained on ImageNet vs. zero-shot language-image pre-trained models trained on LAION. In this paper, we propose a new evaluation metric to evaluate and compare the effective robustness of models trained on different data. To do this, we control for the accuracy on multiple ID test sets that cover the training distributions for all the evaluated models. Our new evaluation metric provides a better estimate of effective robustness when there are models with different training data. It may also explain the surprising effective robustness gains of zero-shot CLIP-like models exhibited in prior works that used ImageNet as the only ID test set, while the gains diminish under our new evaluation. Additional artifacts including interactive visualizations are provided at https://shizhouxing.github.io/effective-robustness.

## 1  Introduction

Robustness against distribution shifts is important for machine learning models to work reliably across various environments. For natural distribution shifts on image classification datasets, Taori et al. (2020) proposed the notion of *effective robustness* to control for in-distribution (ID) accuracy when evaluating out-of-distribution (OOD) accuracy. Following a long line of work that has found a strong correlation between ID and OOD accuracy on many test sets (Recht et al., 2019; Yadav & Bottou, 2019), effective robustness allows researchers to assess whether an apparently improved OOD accuracy is a result of effectively improved robustness or is simply an expected outcome of enhanced ID accuracy.

Unfortunately, the current definition of effective robustness has a subtle limitation: it requires a fixed ID test set, which is typically ImageNet (Deng et al., 2009) when using ImageNet-like OOD test sets

---

[*]Work done while at Google.

37th Conference on Neural Information Processing Systems (NeurIPS 2023).

in Taori et al. (2020) or CIFAR-10 (Krizhevsky et al., 2009) when using CIFAR-like OOD test sets in Miller et al. (2021). It is acceptable when models are trained predominately on only one dataset. However, the emergence of many large-scale models trained on significantly different datasets makes it necessary to evaluate and compare models trained on different data distributions, under which it becomes unclear which ID test set should be used.

In particular, models from Contrastive Language-Image Pre-training, such as CLIP (Radford et al., 2021) and ALIGN (Jia et al., 2021) have recently exhibited unprecedented effective robustness gains during *zero-shot* inference (Radford et al., 2021; Fang et al., 2022; Nguyen et al., 2022). However these previous works simply take ImageNet as the single ID test set, even though the models are not trained on ImageNet. We demonstrate that the results of evaluating effective robustness using a single ID test set can vary drastically depending on the selection of the ID test set. Therefore, this imprecise treatment on the ID test set in existing works could end up exaggerating the effective robustness of zero-shot CLIP models compared to models that are exactly trained on ImageNet.

In this paper, we propose to more precisely evaluate and compare the effective robustness of models trained on different datasets. Instead of controlling for a single ID accuracy that may bias towards models from a particular training distribution, we propose to use multiple ID test sets that cover the training distributions of all the models. In particular, previous works performed single-dimensional linear regression on a set of baseline models to predict OOD accuracy from a single ID accuracy (Taori et al., 2020). And they then evaluate the actual OOD accuracy of the models *beyond* the expected value that can be predicted from the fitting line, as the effective robustness. We expand on this definition by allowing for multiple ID test sets, and perform *multi-dimensional* linear regression to fit a plane to predict OOD accuracy from the accuracy on multiple ID test sets.

In summary, we make the following contributions:

- We reveal a limitation in the existing effective robustness evaluation when used to compare models trained on different data distributions.
- We then propose a new effective robustness evaluation which uses multiple ID test sets to more precisely compare the effective robustness of models trained on different data.
- We show that the OOD accuracy of various models including zero-shot CLIP models can usually be better predicted from accuracies on multiple ID test sets compared to using only one ID test set.
- Our results provide new understandings on the effective robustness gains of CLIP-like models observed in prior works only using ImageNet as the ID test set, while the gains diminish under our new evaluation.

## 2 Background of Effective Robustness

Under natural distribution shifts, the OOD accuracy of a model is often correlated with the ID accuracy. After applying a logit transformation on the accuracy, a linear trend between the transformed ID accuracy and OOD accuracy holds across many datasets (e.g., a distribution shift from ImageNet (Deng et al., 2009) to ImageNetV2 (Recht et al., 2019), or from CIFAR-10 (Krizhevsky et al., 2009) to CIFAR-10.2 (Hendrycks & Dietterich, 2018)) and models with various architectures and training methods (Taori et al., 2020; Miller et al., 2021). This phenomenon implies that most models showing higher OOD accuracies naturally resulted from better ID performance.

To eliminate the confounding effect of ID accuracy on OOD performance, Taori et al. (2020) proposed *effective robustness* that measures the OOD performance *beyond* the expected OOD accuracy given the ID accuracy, where the expected OOD accuracy is predicted according to the fitted linear trend of baseline models. Since they only use a single ID test set, we refer to this version of effective robustness as *single-ID effective robustness*.

Suppose there are $n$ baseline models $f_1, f_2, \cdots f_n$. A baseline function $\tilde{\beta}(x)$ is constructed to predict the OOD accuracy of each baseline model, $\mathrm{acc}_{\mathrm{ood}}(f_i)$ $(1 \leq i \leq n)$, given the single ID accuracy of the model $x = \mathrm{acc}_{\mathrm{id}}(f_i)$. The baseline function is instantiated as:

$$\tilde{\beta}(x) = \mathrm{expit}(w \, \mathrm{logit}(x) + b), \tag{1}$$

where $w$ and $b$ are parameters, $\mathrm{logit}(x) = \ln(\frac{x}{1-x})$ is the logit transformation, and $\mathrm{expit}(x)$ is the inverse of $\mathrm{logit}(x)$. Since $\mathrm{logit}(\tilde{\beta}(x)) = w \, \mathrm{logit}(x) + b$, the baseline function is essentially a linear

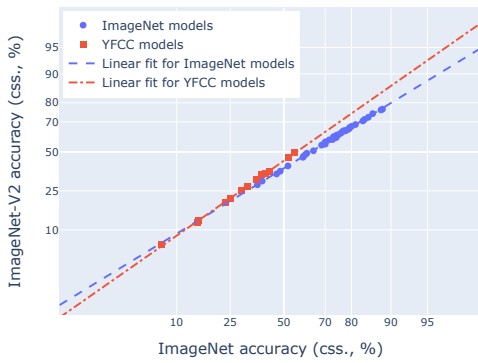 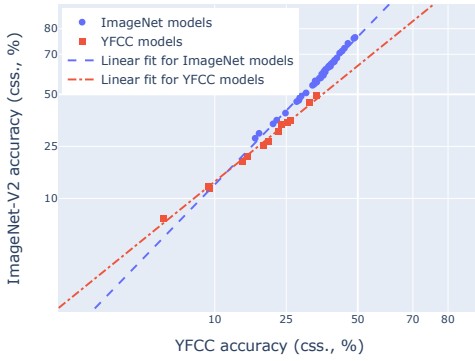

(a) ImageNet-V2 accuracy against ImageNet accuracy.  (b) ImageNet-V2 accuracy against YFCC accuracy.

Figure 1: Class-subsampled ("css." for short) ImageNet-V2 accuracy against ImageNet accuracy and YFCC accuracy, respectively, for 36 ImageNet models and 13 YFCC models that are also used in Table 3a. A linear fit is generated for ImageNet models and YFCC-15M models, respectively. Accuracies and linear fits are under the logit scale. Class-subsampling is used to only include classes that appear in all the involved test sets (see Section 5.1).

function after applying a logit transformation on the accuracies, and it can be determined by solving a linear regression. Then the single-ID effective robustness of a model $f$ is evaluated as

$$\tilde{\rho}(f) = \text{acc}_{\text{ood}}(f) - \tilde{\beta}(\text{acc}_{\text{id}}(f)), \tag{2}$$

which subtracts the predicted OOD accuracy based on the ID accuracy $\text{acc}_{\text{id}}(f)$, from the actual OOD accuracy $\text{acc}_{\text{ood}}(f)$.

## 3 Limitation of the Single ID Test Set

The existing effective robustness evaluation in Section 2 fixes a single ID test set for all the models, which is reflective of the ID performance only if all the models are trained on the same dataset that matches the ID test set. However, as large-scale pre-trained models emerge, it becomes necessary to compare models trained on different datasets, in order to know if the latest pre-training techniques can yield effective robustness gains. In this section, we use the comparison between zero-shot CLIP models and standard ImageNet models as an example to show the limitation of using a single ID test set: when only one ID test set is used, using different ID test sets leads to contradictory conclusions.

Following Fang et al. (2022), we compare models trained on ImageNet (Deng et al., 2009) and YFCC-15M (Thomee et al., 2016), respectively. On ImageNet, we include standard classifiers, and we also train CLIP models using templates filled with an ImageNet class name as the caption in a format of "A photo of a {class name}". We also train CLIP models on YFCC-15M, a dataset with image-text pairs. And we use ImageNet-V2 (Recht et al., 2019) as the OOD test set. We consider two different ID test sets. One ID test set is simply ImageNet. The other ID test set is constructed from YFCC-15M, since we have CLIP models trained on YFCC. We refer to this test set as "YFCC test set", and we refer to the accuracy on this test set as "YFCC accuracy". We discuss its details in Section 5.1 and Appendix B.2. Both ID test sets we consider here match the training distribution of some of the models (ImageNet models and YFCC models respectively) but not all the models.

We then plot the ImageNet-V2 accuracy of the models against their ImageNet accuracy and YFCC accuracy, respectively. There is a strong linear trend between the scaled ID accuracy and OOD accuracy for ImageNet models and YFCC models, respectively, and we plot fitting lines for these two sets of models, respectively. When the ID test set is ImageNet, Fig. 1a shows that the fitting line for YFCC models is generally above the fitting line for ImageNet models (except for the regime with extremely low accuracies), which appears to suggest that YFCC models have effective robustness gains over ImageNet models, as also suggested in Fang et al. (2022). However, in Fig. 1b which uses YFCC as the ID test set, the fitting line of ImageNet models are now mostly above YFCC models, which instead appears to suggest that ImageNet models have greater effective robustness than YFCC models. We observe that when there is a mismatch in the training data and the ID test data, the

models appear to have greater effective robustness (YFCC models in Figure 1a and ImageNet models in Figure 1b), as their performance on the ID test data and the OOD performance predicted from the single ID accuracy tend to be lower. This makes it difficult to compare models trained on different data and leads to imprecise conclusions on effective robustness if only one ID test set is used.

## 4   Multi-ID Effective Robustness

Considering the limitations of using a single ID test set, we propose a new way for effective robustness evaluation using multiple ID test sets that cover the training data distributions of all the involved models. We name it *multi-ID effective robustness*. Specifically, for each training distribution, we propose to prepare an ID test set that matches the training distribution, respectively. In particular, we focus on comparing models trained on two different datasets at a time in this paper, and we thereby use two ID test sets, where each of them corresponds to one of the training datasets.

While we refer to them as ID test sets, each of them is only the exact ID test set for some of the considered models that are trained on the distribution matching the test set, and it is not exactly an ID test set for all the considered models. However, we assume that the training distributions of all the models are still relatively close compared to the OOD test distributions (e.g., images normally collected from social medias in ImageNet (Deng et al., 2009), YFCC (Thomee et al., 2016), and LAION (Schuhmann et al., 2021) are relatively close compared to the OOD images in ImageNet-Sketch (Wang et al., 2019) that consists of sketch images specifically). In this way, both ID test sets are relatively ID for all the models compared to the OOD test sets, and it can be meaningful to control for the performance on these ID test sets when comparing the OOD performance.

We still use $\mathrm{acc_{ood}}(\cdot)$ to denote the OOD accuracy, and we use $\mathrm{acc_1}(\cdot)$ and $\mathrm{acc_2}(\cdot)$ to denote the accuracy on the two ID test sets, respectively. In contrast to the previous baseline function $\tilde{\beta}(x)$ in Eq. (1), we propose a new baseline function $\beta(x, y)$ that predicts the OOD accuracy based on the accuracies $x$ and $y$ on the two ID test sets, respectively.

All the models in Figure 1 are trained on either ImageNet or YFCC. Thus, to compare their effective robustness under our new evaluation, we use two ID test sets for ImageNet and YFCC at the same time, in contrast to Figure 1a and 1b which use one ID test set separately at each time and results on the two different ID test sets lead to contradictory conclusions. As shown in Figure 2, we plot the OOD accuracy against the two ID accuracies on both two ID test sets in a 3D space. We observe that the data points approximately lie on a plane when plotted on the logit scale. This motivates us to instantiate $\beta(x, y)$ as:

$$\beta(x, y) = \mathrm{expit}(w_x \log\mathrm{it}(x) + w_y \log\mathrm{it}(y) + b), \tag{3}$$

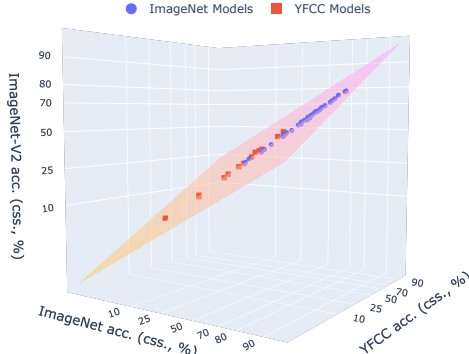

Figure 2: Class-subsampled ("css." for short) ImageNet-V2 accuracy against both ImageNet accuracy and YFCC accuracy for ImageNet models and YFCC models used in Figure 1 which shows the projections when only one of ImageNet accuracy and YFCC accuracy is used.

where $w_x, w_y, b$ are parameters. $\beta(x, y)$, which is the plane in Figure 2, is also a linear function w.r.t. $x$ and $y$ under the logit scale, and thus it is a reasonable extension from $\tilde{\beta}(x)$ by using a multi-dimensional linear function on the logit scale. We determine the parameters by solving an ordinary least squares regression to fit the accuracies. Metrics for linear regression such as the coefficient of determination, a.k.a. $R^2$, can be used to evaluate the fitting quality of the baseline function. A high $R^2$ value indicates that the OOD accuracy is accurately predicted by the baseline function from the ID accuracies, and thus the evaluated models have similar effective robustness. And our multi-ID effective robustness for a model $f$ is defined as

$$\rho(f) = \mathrm{acc_{ood}}(f) - \beta(\mathrm{acc_1}(f), \mathrm{acc_2}(f)).$$

Compared to the existing definition for effective robustness in Eq. (2), the major difference is the inclusion of two ID accuracies $\mathrm{acc_1}(f)$ and $\mathrm{acc_2}(f)$ in the baseline function, compared to using a single ID accuracy $\mathrm{acc_{id}}(f)$.

**Generalizing to more than two training datasets.**   Although we focus on handling two training datasets at a time, our method may be generalized to more than two datasets in principle, by defining a baseline function based on multiple ID accuracies $\text{acc}_1(\cdot), \cdots, \text{acc}_k(\cdot)$. However, it could be costly as it would require training more models to fit a high-quality baseline function. We leave it for future works to reduce the cost when dealing with a larger number of datasets.

## 5   Experiments

### 5.1   Settings

**Models.**   In order to fit a baseline function, we need a large amount of models with diverse accuracies. To this end, we follow Taori et al. (2020) to train models with various proportions of data by subsampling from the entire training set (namely dataset subsampling), which effectively produces models with diverse accuracies. Moreover, we also combine examples from two datasets with different sampling ratios and train models on these combined datasets. This produces models with training distributions varying between the two training datasets and it is supposed to yield different combinations of the two ID accuracies. We use models trained on each single dataset as well as the combined datasets in the same fitting process, so that the baseline functions do not bias towards models trained on certain data. Our experiments include the following models:

- **Standard classifiers on CIFAR-10 and ImageNet.** We train standard classifiers on CIFAR-10 and ImageNet (Deng et al., 2009). We use ResNet-18, ResNet-50, and ResNet-101 (He et al., 2016). Additionally, we train models by combining CIFAR-10 and ImageNet at various ratios, where we upsample CIFAR-10 images from $32 \times 32$ to $224 \times 224$. Furthermore, we include ViT-S/16, ViT-B/16, ViT-L/16 models (Dosovitskiy et al., 2021) pre-trained on the whole ImageNet.
- **CLIP models.** On YFCC-15M (Thomee et al., 2016) and LAION-15M (Schuhmann et al., 2021) which consist of image-text pairs, we train CLIP models using ResNet-50 and ResNet-101. We also train models by combining ImageNet and YFCC-15M and LAION-15M, respectively. We discard models with ImageNet accuracy below 5%. Additionally, in Section 5.4, we also have downloaded ViT-based models from Mu et al. (2022); Ilharco et al. (2021) and CLIP models fine-tuned on ImageNet, which are only used for evaluation but not fitting the baseline functions. We provide additional details in Appendix B.1.

We use "{Name_of_dataset} models" to denote models trained only on the dataset, e.g., "CIFAR-10 models". And we use "{Name_of_dataset_A}+{Name_of_dataset_B} models" to represent models trained on a combination of two datasets, e.g., "CIFAR-10+ImageNet models".

**ID test sets.**   We focus on image classification. Labeled image classification datasets such as ImageNet can be directly used for evaluating ID accuracy. For datasets that consist of image-text pairs for language-image pre-training without original labels, including YFCC and LAION, we automatically generate classification labels by matching captions with ImageNet classes, which has been similarly performed in Fang et al. (2022) for training classifiers using caption data, and we then hold out a balanced test set from the original dataset. More details are reported in Appendix B.2. Although it is possible to obtain a higher-quality test set by human labelling, we will show that using the automatically labelled test sets can already produce reasonable results.

**OOD test sets.**   To compare the effectiveness robustness of models trained on CIFAR-10 and ImageNet, we use 3 CIFAR-like OOD test sets with natural distribution shifts, including CIFAR-10.1 (Recht et al., 2019), CIFAR-10.2 (Lu et al., 2020), and CINIC-10 (Darlow et al., 2018). We use 4 ImageNet-like OOD test sets to compare models trained on ImageNet with models trained on YFCC and LAION: ImageNet-V2 (Recht et al., 2019), ImageNet-R (Hendrycks et al., 2021a), ImageNet-Sketch (Wang et al., 2019), and ObjectNet (Barbu et al., 2019). We do not use ImageNet-A (Hendrycks et al., 2021b) which involves adversarial filtering and has a different behavior in effective robustness evaluation (Taori et al., 2020; Fang et al., 2022).

**Class subsampling and mapping.**   Considering that different test sets may not have the same classes, we follow prior works (Taori et al., 2020; Fang et al., 2022) to use class subsampling[2] to

---

[2]We reuse the term "class subsampling" from prior works (Taori et al., 2020; Fang et al., 2022), although it is not a random sampling.

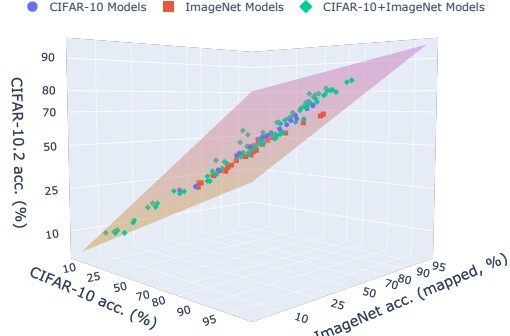
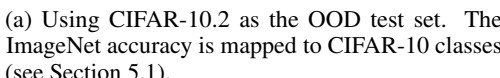
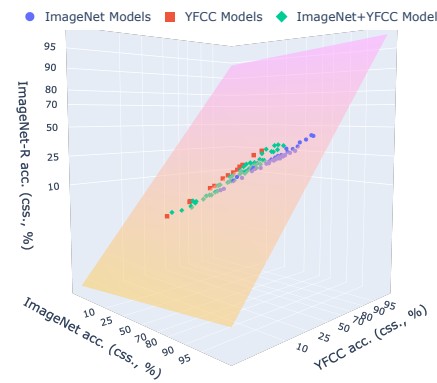

(a) Using CIFAR-10.2 as the OOD test set. The ImageNet accuracy is mapped to CIFAR-10 classes (see Section 5.1).

(b) Using ImageNet-R as the OOD test set.

Figure 3: Visualization of the multi-ID effective robustness. The colored plane stands for the baseline function. Figure 4 and Figure 5 (in Appendix A.1) show various projected 2D views. See our website (https://shizhouxing.github.io/effective-robustness) for an interactive visualization.

retain classes appearing in all the test sets. We also follow Miller et al. (2021) to map a subset of ImageNet classes to CIFAR-10 classes when comparing CIFAR-10 models and ImageNet models,.

## 5.2 Evaluation on CIFAR-like OOD Test Sets

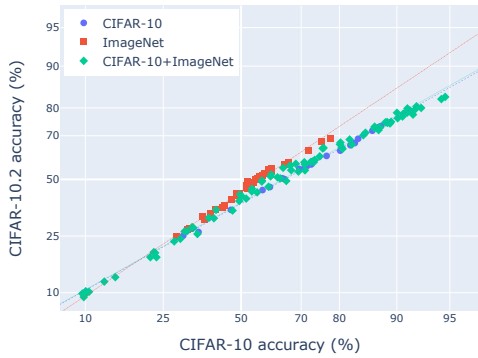
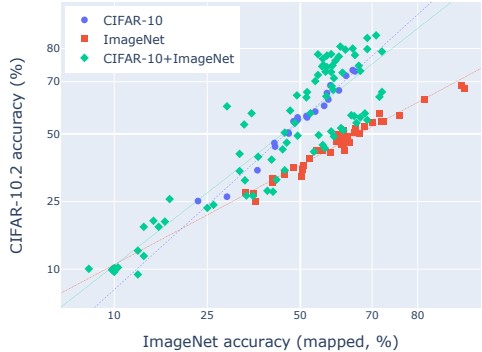

(a) CIFAR-10.2 accuracy against CIFAR-10 accuracy. ImageNet models have higher CIFAR-10.2 accuracy compared to CIFAR-10 models when controlling for CIFAR-10 accuracy only.

(b) CIFAR-10.2 accuracy against ImageNet accuracy. ImageNet models have lower CIFAR-10.2 accuracy compared to CIFAR-10 models when controlling for ImageNet accuracy only.

Figure 4: Projected views of Figure 3a. Figure 4a and Figure 4b correspond to single-ID evaluations using different ID test sets and yield contradictory conclusions on the effective robustness. Our multi-ID evaluation provides a more holistic view where all these models are approximately on a same plane and thus have similar effective robustness.

We first experiment with models trained using CIFAR-10 and ImageNet on CIFAR-like OOD test sets. We show the fitting quality in Table 1a and the effective robustness of various models in Table 1b. Compared to the single-ID evaluation, our multi-ID evaluation achieves a better fitting quality and predicts the OOD accuracy from the ID accuracies more precisely (higher $R^2$ and lower MAE), and thus provides a more precise understanding on the effective robustness. Specifically, while both single-ID effective robustness and multi-ID effective robustness have relatively high fitting quality on CIFAR-like test sets, using multi-ID effective robustness further improves the fitting quality. In terms of the effective robustness, under the single-ID evaluation, ImageNet models achieve 3.91±2.20 (%) and 2.77±1.25 (%) effective robustness on CIFAR-10.2 and CINIC-10, respectively. The positive

Table 1: Results on CIFAR-like OOD test sets. 148 models are included, including CIFAR-10 models, ImageNet models, and CIFAR-10+ImageNet models (CIFAR+IN for short). The multi-ID evaluation achieves better fitting quality where the effective robustness values of CIFAR-10 models and ImageNet models also become closer to 0.

(a) Fitting quality evaluated by $R^2$ and mean absolute error (MAE).

| Test set | $R^2$ ($\uparrow$) | | MAE (%, $\downarrow$) | |
|---|---|---|---|---|
| | Single-ID | Multi-ID | Single-ID | Multi-ID |
| CIFAR-10.1 | 0.996 | **0.997** | 1.07 | **0.93** |
| CIFAR-10.2 | 0.981 | **0.996** | 2.22 | **0.95** |
| CINIC-10 | 0.978 | **0.990** | 2.41 | **1.49** |

(b) Effective robustness values (%). We report the mean and standard deviation for three groups of models with different training data, respectively.

| Test set | Evaluation | CIFAR-10 21 models | ImageNet 89 models | CIFAR+IN 38 models |
|---|---|---|---|---|
| CIFAR-10.1 | Single-ID | -1.68±0.92 | 1.05±1.27 | 0.02±1.10 |
| | Multi-ID | -1.43±0.92 | 0.10±1.12 | 0.19±1.01 |
| CIFAR-10.2 | Single-ID | -1.65±0.70 | 3.91±2.20 | -0.64±1.79 |
| | Multi-ID | -0.76±0.77 | 0.56±1.27 | 0.03±1.29 |
| CINIC-10 | Single-ID | -0.96±1.43 | 2.77±1.25 | -0.10±2.81 |
| | Multi-ID | -0.08±1.52 | -0.52±0.98 | 0.63±2.10 |
| Average | Single-ID | -1.43±0.53 | 2.58±1.32 | **-0.24±1.58** |
| | Multi-ID | **-0.76±0.63** | **0.04±0.67** | 0.28±1.04 |

effective robustness values seems to suggest an advantage of ImageNet models compared to CIFAR-10 models, which is consistent with the findings in Miller et al. (2021). However, under the multi-ID evaluation, the advantage of ImageNet models diminishes, and the effective robustness values of both CIFAR-10 models and ImageNet models are much closer to 0. Therefore, the apparent advantage reported by prior works can be explained as the effect of training data on the single-ID evaluation, and our multi-ID evaluation resolves this confounder to provide a more precise understanding.

In Figure 3a, we visualize the multi-ID effective robustness on CIFAR-10.2, where the accuracies of all the models approximately lie on a plane (the baseline function) on the logit scale, and thus these models have similar effective robustness as the OOD accuracy of all the models can be approximately predicted using a simple plane. We also show projected views of Figure 3a in Figure 4, where Figure 4a and Figure 4b correspond to the single-ID evaluation taking different ID test sets with contradictory conclusions. In contrast, our new evaluation provides a more holistic view.

## 5.3 Evaluation on ImageNet-like OOD Test Sets

We then conduct evaluation on ImageNet-like OOD test sets, and we compare ImageNet models with models trained on YFCC and LAION, respectively. We show the fitting quality in Table 2 and the effective robustness in Tables 3a and 3b. Consistent with results in Section 5.2, our multi-ID evaluation improves the fitting quality over the single-ID evaluation to better predict and understand the OOD accuracies from ID accuracies. On effective robustness, single-ID evaluation leads to a perception of an effective robustness gain when there is mismatch between the training data and the single ID test set. Our multi-ID evaluation enjoys a holistic view of all the training distributions and suggests that all the models evaluated here have similar effective robustness.

Specifically, the improvement of fitting quality is particularly significant for models involving LAION on ImageNet-R ($R^2$ improved from 0.216 to 0.982 and MAE reduced from 9.23% to 1.32%) and ImageNet-Sketch ($R^2$ improved from 0.281 to 0.937 and MAE reduced from 7.90% to 2.10%). On the effective robustness values, under the single-ID evaluation, YFCC and LAION models have positive effective robustness values (2.59±2.43 (%) on average for YFCC models and 5.96±4.96 (%) on average for LAION models), which is consistent with the findings in Fang et al. (2022); Nguyen et al. (2022). In contrast, under our multi-ID evaluation, the average effective robustness becomes 0.77±0.85 (%) for YFCC models, and -0.00±0.52 (%) for LAION models, much closer to 0. While single-ID evaluation used by prior works (Fang et al., 2022; Nguyen et al., 2022) suggests effective

Table 2: Fitting quality of single-ID and multi-ID effective robustness, respectively, on ImageNet-like OOD test sets. For ImageNet v.s. YFCC, models involved include ImageNet, YFCC, and ImageNet+YFCC models. For ImageNet v.s. LAION, models involved include ImageNet, LAION, and ImageNet+LAION models. Multi-ID evaluation improves the fitting quality.

| ImageNet v.s. | Test set | $R^2$ (↑) | | MAE (%, ↓) | |
|---|---|---|---|---|---|
| | | Single-ID | Multi-ID | Single-ID | Multi-ID |
| YFCC (103 models) | ImageNet-V2 | 0.990 | **0.999** | 1.44 | **0.54** |
| | ImageNet-R | 0.879 | **0.965** | 2.55 | **1.30** |
| | ImageNet-Sketch | 0.928 | **0.945** | 1.56 | **1.31** |
| | ObjectNet | 0.903 | **0.936** | 2.60 | **1.98** |
| LAION (107 models) | ImageNet-V2 | 0.992 | **0.999** | 1.33 | **0.51** |
| | ImageNet-R | 0.216 | **0.982** | 9.23 | **1.32** |
| | ImageNet-Sketch | 0.281 | **0.937** | 7.90 | **2.10** |
| | ObjectNet | 0.849 | **0.906** | 2.88 | **2.38** |

Table 3: Single-ID and multi-ID effective robustness (%) of the models on variants of ImageNet OOD test sets. The effective robustness of all the models becomes close to 0 under the multi-ID evaluation.

(a) Models involving ImageNet and YFCC.

| Models | Test set | Single-ID | Multi-ID |
|---|---|---|---|
| ImageNet (36 models) | ImageNet-V2 | -1.23±0.46 | -0.19±0.50 |
| | ImageNet-R | -2.80±1.34 | -0.41±1.83 |
| | ImageNet-Sketch | -1.25±1.90 | 0.14±2.57 |
| | ObjectNet | -0.99±4.23 | 0.74±4.14 |
| | Average | -1.57±1.20 | **0.07±1.68** |
| YFCC (13 models) | ImageNet-V2 | 1.69±1.84 | -0.16±0.57 |
| | ImageNet-R | 3.44±3.25 | 1.07±1.17 |
| | ImageNet-Sketch | 1.90±2.25 | 0.89±1.29 |
| | ObjectNet | 3.32±2.55 | 1.27±0.85 |
| | Average | 2.59±2.43 | **0.77±0.85** |
| ImageNet+YFCC (54 models) | ImageNet-V2 | 0.70±1.75 | 0.13±0.79 |
| | ImageNet-R | 1.31±2.75 | 0.16±1.76 |
| | ImageNet-Sketch | 0.69±1.89 | 0.21±1.52 |
| | ObjectNet | 0.21±1.98 | -0.60±1.01 |
| | Average | 0.73±1.96 | **-0.03±1.06** |

(b) Models involving ImageNet and LAION.

| Models | Test set | Single-ID | Multi-ID |
|---|---|---|---|
| ImageNet (37 models) | ImageNet-V2 | -1.21±0.56 | 0.05±0.65 |
| | ImageNet-R | -9.45±2.79 | -0.54±1.90 |
| | ImageNet-Sketch | -7.63±3.40 | -0.72±3.03 |
| | ObjectNet | -1.90±4.48 | 1.14±4.18 |
| | Average | -5.05±2.21 | **-0.02±1.53** |
| LAION (14 models) | ImageNet-V2 | 1.42±1.73 | -0.03±0.57 |
| | ImageNet-R | 9.48±8.84 | -0.65±1.05 |
| | ImageNet-Sketch | 8.71±7.15 | -1.10±1.98 |
| | ObjectNet | 4.24±2.39 | 1.77±1.20 |
| | Average | 5.96±4.96 | **-0.00±0.52** |
| ImageNet+LAION (56 models) | ImageNet-V2 | 0.65±1.43 | 0.07±0.61 |
| | ImageNet-R | 6.01±8.43 | 0.56±1.32 |
| | ImageNet-Sketch | 5.99±7.39 | 1.04±2.46 |
| | ObjectNet | 0.63±2.20 | -1.01±1.44 |
| | Average | 3.32±4.63 | **0.16±0.65** |

robustness gains of YFCC models compared to ImageNet models (Figure 5a), all the models have similar effective robustness under our multi-ID evaluation. Additionally, we provide an ablation study on using models with mixed training data in Appendix A.3 and additional interactive visualization on our website at `https://shizhouxing.github.io/effective-robustness`.

## 5.4 Evaluation on Additional Models

We also evaluate additional models that are not used in fitting the baseline functions. We download models pre-trained by existing works, including OpenCLIP (Ilharco et al., 2021) and SLIP (Mu et al., 2022). OpenCLIP provides CLIP models trained on YFCC and LAION, and SLIP provides YFCC models trained using CLIP and also a combination of self-supervised learning (Chen et al., 2020a,b) and CLIP (SimCLR+CLIP namely SLIP). And we also fine-tune CLIP models on ImageNet for models we pre-train on YFCC and LAION. We use both vanilla fine-tuning and also WiSE-FT (Wortsman et al., 2022b) which aims to improve the robustness after fine-tuning, using a weight-space ensembling of the pre-trained model and the fine-tuned model. Details are in Appendix B.1.

In Table 4, we show results involving YFCC and LAION, respectively. Since these models are not used in the fitting, we do not use $R^2$, but we use MAE to measure the fitting quality. Our multi-ID evaluation generally reduces MAE compared to the single-ID evaluation, and thus the multi-ID evaluation can still more accurately predict the OOD accuracy from the ID accuracies for these models that are not used in the fitting. The effective robustness values of the models also generally become closer to 0, especially for the zero-shot CLIP models. The results further validate that zero-shot CLIP models, although may achieve high OOD performance if pre-trained with large-scale data (Radford et al., 2021), generally do not improve effective robustness if we control for all the ID accuracies. Among the models evaluated here, SLIP models on YFCC and WiSE-FT models from LAION achieve relatively higher average effective robustness compared to other models, under our multi-ID evaluation, although the gains are not consistently significant on all the test sets and become much smaller than those reflected in the single-ID evaluation. However, we are not certain

Table 4: Fitting quality and effective robustness for downloaded and fine-tuned models. The models are not used in the fitting and directly evaluated. Note that MAE and effective robustness are different, where MAE takes absolute values but not effective robustness. For CLIP by Mu et al. (2022) and SLIP, only models pre-trained on YFCC are available.

| Model | Test set | Models with pre-training on YFCC | | | | Models with pre-training on LAION | | | |
|---|---|---|---|---|---|---|---|---|---|
| | | MAE (%, ↓) | | Effective Robustness (%) | | MAE (%, ↓) | | Effective Robustness (%) | |
| | | Single-ID | Multi-ID | Single-ID | Multi-ID | Single-ID | Multi-ID | Single-ID | Multi-ID |
| OpenCLIP | ImageNetN-V2 | 3.95 | 0.45 | 3.95±0.70 | -0.33±0.45 | 4.70 | 0.38 | 4.70±0.01 | 0.38±0.01 |
| | ImageNetN-R | 8.98 | 3.12 | 8.98±1.24 | 3.12±1.27 | 39.80 | 7.49 | 39.80±0.03 | 7.49±0.09 |
| | ImageNetN-Sketch | 5.32 | 2.49 | 5.32±1.07 | 2.49±0.81 | 38.92 | 1.31 | 38.92±0.00 | -1.31±0.23 |
| | ObjectNet | 5.68 | 1.04 | 5.68±0.81 | -0.22±1.04 | 6.94 | 1.35 | 6.94±0.09 | -1.35±0.04 |
| | Average | 5.98 | 1.77 | 5.98±0.96 | 1.26±0.89 | 22.59 | **2.63** | 22.59±0.02 | **1.30±0.07** |
| Vanilla FT | ImageNetN-V2 | 1.14 | 0.85 | 0.16±1.23 | 0.77±0.70 | 0.82 | 0.91 | -0.12±0.89 | 0.51±0.96 |
| | ImageNetN-R | 2.90 | 1.82 | -2.90±1.86 | -1.82±0.92 | 4.63 | 2.38 | -4.45±2.89 | 2.27±2.20 |
| | ImageNetN-Sketch | 2.26 | 3.16 | 2.20±1.86 | 3.16±1.19 | 7.50 | 7.50 | 3.47±4.67 | 7.50±3.29 |
| | ObjectNet | 3.46 | 3.11 | -3.42±2.36 | -3.11±1.43 | 4.07 | 2.27 | -4.07±2.10 | -2.26±1.95 |
| | Average | 2.44 | **2.23** | -0.99±1.76 | **-0.25±0.96** | 3.57 | **3.27** | -1.29±2.19 | **2.01±1.38** |
| WiSE-FT | ImageNetN-V2 | 1.97 | 1.05 | 1.97±0.76 | 1.05±0.59 | 1.72 | 0.81 | 1.72±0.45 | 0.81±0.48 |
| | ImageNetN-R | 3.64 | 1.76 | 3.64±0.61 | 1.76±0.60 | 13.08 | 7.40 | 13.08±1.85 | 7.40±1.10 |
| | ImageNetN-Sketch | 5.47 | 4.41 | 5.47±1.20 | 4.41±0.99 | 16.84 | 10.21 | 16.84±1.58 | 10.21±0.49 |
| | ObjectNet | 2.12 | 1.30 | 2.12±1.38 | -0.68±1.34 | 1.65 | 1.65 | 1.52±1.64 | 0.31±1.67 |
| | Average | 3.30 | **2.13** | 3.30±0.96 | **1.63±0.80** | 8.32 | **5.02** | 8.29±1.04 | **4.68±0.55** |
| CLIP by Mu et al. (2022) | ImageNetN-V2 | 4.95 | 0.83 | 4.95±0.29 | -0.83±0.45 | | | | |
| | ImageNetN-R | 6.54 | 1.86 | 6.54±2.66 | -1.86±1.15 | | | | |
| | ImageNetN-Sketch | 1.49 | 4.16 | 0.58±1.68 | -4.16±0.44 | | | | |
| | ObjectNet | 9.32 | 1.49 | 9.32±1.45 | 1.49±0.44 | | | | |
| | Average | 5.57 | **2.09** | 5.35±1.49 | **-1.34±0.28** | | - | | |
| SLIP | ImageNetN-V2 | 5.25 | 0.64 | 5.25±0.57 | -0.12±0.84 | | | | |
| | ImageNetN-R | 14.47 | 6.13 | 14.47±5.98 | 5.75±4.77 | | | | |
| | ImageNetN-Sketch | 7.71 | 2.90 | 7.71±4.05 | 2.11±2.79 | | | | |
| | ObjectNet | 14.79 | 5.02 | 14.79±2.77 | 5.02±1.53 | | | | |
| | Average | 10.56 | **3.67** | 10.56±3.11 | **3.19±2.05** | | | | |

on whether SLIP and WiSE-FT can alter the underlying training distribution, because SLIP uses SimCLR that introduces different training images, and WiSE-FT esseentially edits the weights of the models by weight-space ensembling. Thus, we do not draw a definite conclusion for the effective robustness of SLIP and WiSE-FT models and leave further validation for future work.

# 6 Related Work

For natural distribution shifts, the linear correlations between the ID and OOD performance ("accuracy-on-the-line" (Miller et al., 2021) in the single-ID effective robustness) have earlier been observed in dataset reproduction works (Recht et al., 2018, 2019; Yadav & Bottou, 2019; Miller et al., 2020). Taori et al. (2020) evaluated many ImageNet models on several ImageNet-like OOD test sets, and given the widely held linear correlations, they proposed to evaluate effective robustness by controlling for ID accuracy. Miller et al. (2021) further validated accuracy-on-the-line with a broader scope. Nevertheless, accuracy-on-the-line may not hold on some distribution shifts, such as some corruption shifts (Hendrycks & Dietterich, 2018) and shifts in the wild (Miller et al., 2021), and sometimes ID accuracy and OOD accuracy can inversely correlate (Miller et al., 2021; Teney et al., 2022). Baek et al. (2022) also observed a linear correlation between ID agreement and OOD agreement for a pair of neural networks (namely "agreement-on-the-line") for testing whether accuracy-on-the-line holds by agreement-on-the-line which does not require labeled data. We focus on distribution shifts where at least accuracy-on-the-line holds for models from each of the training datasets, and we further propose "accuracy-on-the-plane" using multiple ID test sets.

Recently, CLIP-like models with language image pre-training which has been studied earlier in works such as Sariyildiz et al. (2020); Zhang et al. (2022); Desai & Johnson (2021), were shown to achieve exceptional effective robustness in the single-ID evaluation (Radford et al., 2021; Jia et al., 2021). Fang et al. (2022) analyzed the cause of the effective robustness gain of CLIP and concluded that the pre-training data determined the robustness. Nguyen et al. (2022) experimented on more pre-training data, and they observed difference in the single-ID effective robustness of models trained on different data. While Fang et al. (2022); Nguyen et al. (2022) both suggested that the pre-training data could determine the effective robustness gains, our evaluation suggests that zero-shot CLIP models do not have effective robustness gains. Besides, Kumar et al. (2021); Andreassen et al. (2022); Wortsman

et al. (2022b) studied the robustness of fine-tuned CLIP models. Moreover, Devillers et al. (2021); Santurkar et al. (2022) studied the transfer performance of CLIP models, which is out of our scope on the robustness against natural distribution shifts.

# 7   Conclusion

To conclude, we propose a new and more precise effective robustness evaluation for models with different training data. In our evaluation, the OOD accuracy can generally be better predicted from multiple ID accuracies compared to previous effective robustness evaluation with a single ID test. We find that zero-shot CLIP models pre-trained on language-image data do not have better effective robustness compared to standard image classifiers, and we provide a new understanding of the apparently significant effective robustness gains observed by prior works.

# 8   Limitations and Future Work

There remain several limitations that may be addressed in future works:

- Our method requires fully knowing the training distributions of all the evaluated models, which is not directly applicable for large-scale pre-trained models with private training data. And this also requires the training methods not to significantly alter the training distribution beyond basic data augmentation, while some methods such as SLIP may alter training distributions more significantly, and it is unclear how we can precisely define the training distribution for a model with post-training processing, such as WiSE-FT (Wortsman et al., 2022b) and Model Soups (Wortsman et al., 2022a) with weight-space ensembling. Future work may study how these techniques may impact the ID performance evaluation (Section 5.4).

- We assume that the two ID test sets have relatively close distributions compare to the OOD test sets. We have not considered how the difference between the multiple ID test sets may affect the evaluation, and how the effective robustness should be compared if models are trained on highly different distributions.

- We use fixed OOD test sets to evaluate the OOD performance, following previous works (Radford et al., 2021; Fang et al., 2022; Nguyen et al., 2022; Schuhmann et al., 2021). When models are pre-trained on large-scale data, it becomes unclear if these "OOD" test sets are still OOD, or if these test sets could be less OOD for the pre-trained models compared to standard classifiers. There may also be some distribution overlap between these test sets and the pre-training datasets, even though Radford et al. (2021) mentioned that the likelihood of direct data overlapping is low.

- We focus on distribution shifts where at least "accuracy-on-the-line" from existing works is known to hold for models trained on the same data (Taori et al., 2020; Miller et al., 2021), yet there are counterexamples where "accuracy-on-the-line" does not hold (Section 6) and requires further study. We may set a threshold on the fitting quality and only adopt our method when the fitting quality is sufficiently good. And there are also other OOD test sets (Singla & Feizi, 2021; Rusak et al., 2021; Singla et al., 2022; Idrissi et al., 2022; Li et al., 2023; Moayeri et al., 2022; Vasudevan et al., 2022) that have not been studied in the effective robustness works yet.

- While we mostly focus on comparing CLIP-like models with standard image classifiers, due to the notable OOD robustness of CLIP-like models studied in prior works (Radford et al., 2021; Fang et al., 2022; Nguyen et al., 2022), this work may further be extended to cover other types of models (Goyal et al., 2022; Singh et al., 2022) as well as other modalities such as distribution shifts on language data (Miller et al., 2020; Awadalla et al., 2022).

- Our multi-ID evaluation is intended for the scenario with models trained on different data. For models trained on a single dataset, while there is often a correlation between the rankings derived from the single-ID evaluation and the multi-ID evaluation, respectively, the rankings are not necessarily consistent (see Appendix A.2), and thus our multi-ID evaluation is not intended to replace the single-ID evaluation in this case. We suggest using single-ID and multi-ID evaluation comprehensively.

## Acknowledgments & Funding Disclosure

We thank Alex Fang and Jindong Gu for helpful discussions and the reviewers for constructive feedback. This work was supported in part by NSF 2008173, 2048280, 2325121, 2331966, ONR N00014-23-1-2300:P00001.

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

# A  Additional Results

## A.1  Projected Views

In Figure 5, we show projected views of Figure 3b.

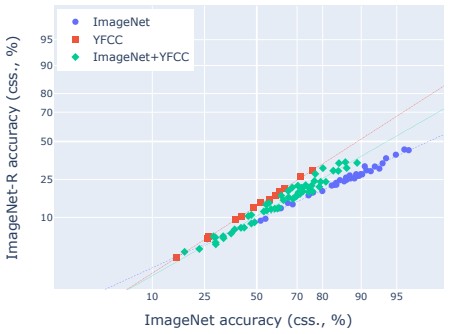 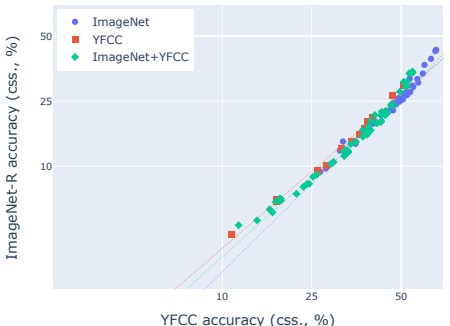

(a) ImageNet-R accuracy against ImageNet accuracy. YFCC models have higher ImageNet-R accuracy compared to ImageNet models when controlling for ImageNet accuracy only.

(b) ImageNet-R accuracy against YFCC accuracy. YFCC models have similar ImageNet-R accuracy compared to ImageNet models when controlling for YFCC accuracy only.

Figure 5: Projected views of Figure 3b. Figure 5a and Figure 5b correspond to single-ID evaluation using diffrent ID test sets, Figure 5a suggests effective robustness gains of YFCC models but the gains diminish in Figure 5b. Our multi-ID evaluation shows a holistic view where all the models have a similar effective robustness.

## A.2  Agreement between Single-ID and Multi-ID Evaluation

We conduct an experiment to check the correlation between the single-ID evaluation and our new multi-ID evaluation, in terms of the relative ranking between different models trained on the same data. We use Kendall's rank correlation test (Kendall, 1948) and we report the $\tau$ statistic computed by `scipy.stats.kendalltau` in Tables 5 to 7. Results show that the rankings on the single-ID effective robustness and multi-ID effective robustness are positively correlated for CIFAR-10 and ImageNet models. There is also a weaker positive correlation for YFCC models. For LAION models, there is sometimes a negative correlation. Overall, while there is often a positive correlation between the rankings provided by the single-ID evaluation and the multi-ID evaluation, respectively, it is not necessarily consistent on all the datasets. Thus, when comparing models trained on the same data, our multi-ID evaluation is not intended to replace the single-ID evaluation. Our multi-ID evaluation is mainly for comparing models trained on different data, and may be used as a supplementary evaluation if all the models are trained on a single dataset.

Table 5: $\tau$ statistics in the Kendall's rank correlation test for evaluating the correlation between the rankings provided by the single-ID evaluation and the multi-ID evaluation, respectively, for models trained on the same data. We consider CIFAR-10 models and ImageNet models, respectively, on CIFAR-like OOD test sets. For ImageNet models, ImageNet instead of CIFAR-10 is used as the ID test set in the single-ID evaluation.

| Test set | CIFAR-10.1 | CIFAR-10.2 | CINIC-10 |
|---|---|---|---|
| CIFAR-10 models | 0.9619 | 0.6667 | 0.8095 |
| ImageNet models | 0.1664 | 0.1522 | 0.4907 |

## A.3  Models with Mixed Training Data in the Fitting

As mentioned in Section 5.1, we train models with mixed data to obtain models with diverse accuracies. In Tables 8 and 9, we show that if we do not include models with mixed training data in the fitting, the MAE for these models can become higher, although the difference is not large.

Table 6: $\tau$ statistics (similar to Table 5) for ImageNet models and YFCC models on ImageNet-like OOD test sets. For YFCC models, YFCC instead of ImageNet is used as the ID test set in the single-ID evaluation.

| Test set | ImageNet-V2 | ImageNet-R | ImageNet-Sketch | ObjectNet |
|---|---|---|---|---|
| ImageNet models | 0.5142 | 0.4412 | 0.8031 | 0.8063 |
| YFCC models | 0.0256 | 0.8461 | 0.7179 | 0.2564 |

Table 7: $\tau$ statistics (similar to Table 5) for ImageNet models and LAION models on ImageNet-like OOD test sets. For LAION models, LAION instead of ImageNet is used as the ID test set in the single-ID evaluation.

| Test set | ImageNet-V2 | ImageNet-R | ImageNet-Sketch | ObjectNet |
|---|---|---|---|---|
| ImageNet models | 0.3123 | 0.4384 | 0.6216 | 0.9729 |
| LAION models | -0.4945 | 0.6483 | -0.2747 | 0.6483 |

In this work, we train models with data mixed at various ratios and consider models with diverse combinations of accuracies, to obtain more convincing conclusions on the effective robustness.

Table 8: MAE (%) for ImageNet+YFCC models when they are excluded and included in the fitting, respectively, where comparing the effective robustness of ImageNet models and YFCC models.

| Test set | ImageNet+YFCC models in the fitting | |
|---|---|---|
| | Excluded | Included |
| ImageNet-V2 | 0.71 | 0.64 |
| ImageNet-R | 1.30 | 1.21 |
| ImageNet-Sketch | 0.98 | 0.94 |
| ObjectNet | 1.73 | 0.95 |

# B   Experimental Details

## B.1   Details of Models

We use TF-Vision[3] under the Apache License Version 2.0 to train standard classifiers on CIFAR-10 and ImageNet. We follow the configurations provided in TF-Vision for vanilla ResNet training on ImageNet and we train ResNet-18, ResNet-50 and ResNet-101 models. We reuse the configurations to train models on CIFAR-10, where we only change the dataset, number of classes, and image size, without tuning hyperparameters for the training. And we load checkpoints of ViT-S/16, ViT-B/16, and ViT-L/16 models pre-trained on ImageNet, provided by TF-Vision.

For training CLIP models, we mostly follow hyperparameters provided in Fang et al. (2022) and the implementation in Open-CLIP (Ilharco et al., 2021). While Fang et al. (2022) used a batch size of 1024, we use 2048 for more parallelism. We use YFCC-15M in Radford et al. (2021), which is a subset of YFCC-100M (Thomee et al., 2016). And we use LAION-15M which we uniformly sample from LAION-400M (Schuhmann et al., 2021). For fine-tuning CLIP models, we fine-tune for 50,000 steps, using learning rates $3 \times 10^{-5}$ and $1 \times 10^{-4}$, respectively. For WiSE-FT, we take $\alpha = 0.5$ which is the coefficient for weight-space ensembling. For OpenCLIP models[4], we use ViT-B/32 models trained on LAION-400M. For SLIP [5], we use all the CLIP and SLIP models trained on YFCC-15M.

For data subsampling, we uniformly sample a proportion of training examples from the entire dataset, at ratios of $\{5\%, 10\%, 20\%, 30\%, 40\%, 50\%\}$, respectively. For combining two training datasets at various ratios, given a coefficient $\lambda$ $(0 < \lambda < 1)$, we uniformly sample a proportion of data from the two datasets at ratios of $\lambda$ and $(1 - \lambda)$, respectively, and then we combine the two subsets. When combining ImageNet and CIFAR-10, we take $\lambda \in \{0.001, 0.01, 0.1, 0.5, 0.9, 0.99, 0.995\}$; when combining ImageNet with YFCC and LAION, respectively, we take $\lambda \in \{0.01, 0.1, 0.25, 0.5\}$.

---

[3]https://github.com/tensorflow/models/tree/master/official/vision
[4]https://github.com/mlfoundations/open_clip
[5]https://github.com/facebookresearch/SLIP

Table 9: MAE (%) for ImageNet+LAION models when they are excluded and included in the fitting, respectively, where comparing the effective robustness of ImageNet models and LAION models.

| Test set | ImageNet+LAION models in the fitting | |
|---|---|---|
| | Excluded | Included |
| ImageNet-V2 | 0.54 | 0.51 |
| ImageNet-R | 1.56 | 1.19 |
| ImageNet-Sketch | 2.62 | 2.03 |
| ObjectNet | 2.91 | 1.47 |

Models are trained using $4 \times 4$ or $4 \times 8$ TPU v2 Pods, and they are evaluated using NVIDIA V100 GPUs, on the cloud.

## B.2  Details of ID Test Sets

We construct an ID test set from YFCC-15M and LAION-15M, respectively, and we automatically generate classification labels by matching text with ImageNet class names, which has also been similarly performed in Fang et al. (2022) for training classifiers using caption data. On YFCC-15M which contains metadata, we use tags for matching. On LAION-15M which does not provide metadata such as tags, we simply use the entire text for matching. We adopt a label only if a unique ImageNet class can be determined by matching for the involved image. We then construct a balanced and labelled test set by keeping 50 examples for each class that has at least 100 examples in the labelled images. The test examples are then held out from the training data. For the YFCC test set, there are 22550 examples for 451 classes; and for the LAION test set, there are 20400 examples 408 classes.

## B.3  Licenses of the Datasets

We have used the following datasets:

- CIFAR-10 (Krizhevsky et al., 2009). License is not clearly known.
- YFCC[6] under various Creative Commons licenses.
- LAION[7] under the Creative Common CC-BY 4.0 license for the metadata, and the images are under the copyright of the original authors.
- CIFAR-10.1[8] under the MIT license.
- CIFAR-10.2[9]. License is not clearly known.
- CINIC-10[10] under the MIT license.
- ImageNet-V2[11] under the MIT license.
- ImageNet-R[12] under the MIT license.
- ImageNet-Sketch[13] under the MIT license.
- ObjectNet[14] with a license provided on the webpage.

---

[6]https://multimediacommons.wordpress.com/yfcc100m-core-dataset
[7]https://laion.ai/blog/laion-400-open-dataset
[8]https://github.com/modestyachts/CIFAR-10.1
[9]https://github.com/modestyachts/cifar-10.2
[10]https://github.com/BayesWatch/cinic-10
[11]https://github.com/modestyachts/ImageNetV2
[12]https://github.com/hendrycks/imagenet-r
[13]https://github.com/HaohanWang/ImageNet-Sketch
[14]https://objectnet.dev/download.html

