# OpenReview forum: "Effective Robustness against Natural Distribution Shifts for Models with Different Training Data"
_NeurIPS.cc/2023/Conference — NeurIPS 2023 poster_

### Official Review · Reviewer_MjfW · 2023-07-04

**Soundness:** 4 excellent
**Presentation:** 3 good
**Contribution:** 3 good
**Rating:** 7
**Confidence:** 5

**Summary:**

The authors propose a modification to effective robustness evaluations that takes multiple training sets into account. This facilitates a more precise comparison of the effective robustness of models trained on different training sets. The authors demonstrate empirically that their method yields an improved predictor of OOD accuracy. The analysis of models trained on ImageNet, YFCC, and LAION reveals that CLIP models might not, in general, exhibit larger effective robustness than ResNets, further strengthening the existing empirical evidence in this direction.

**Strengths:**


The paper is well written and easy to follow. In more detail:
- The introduction is very well written and motivates the ideas clearly. The problem targeted in the paper makes sense.
- The motivating experiment in Section 3 is very nice and illustrates the limitation of using a single ID test set very well and the provided intuition in line 104 makes sense (“We observe that when there is a mismatch in the training data and the ID test data, the models appear to have greater effective robustness (YFCC models in Figure 1a and ImageNet models in Figure 1b), as their performance on the ID test data and the OOD performance predicted from the single ID accuracy tend to be lower.”)


The issue that the regular definition of effective robustness is confounded by the training distribution makes sense and makes it difficult to compare the performance of models trained on different datasets. The idea of doing regression on the accuracies of multiple ID test sets is simple and seems to work well. The conclusion that CLIP models do, in general, not exhibit greater effective robustness is important and relevant.


**Weaknesses:**

While the paper is generally well written, there is an abundance of typos and grammar errors (see examples below), which need fixing before acceptance. I would strongly suggest to the authors to do a round of proof-reading the manuscript.


L40: “In particular, previous works performed single-dimensional linear regression on a set of baseline models to predict OOD accuracy from a single ID accuracy.” citations needed


Line 58: “Under natural distribution shifts, the OOD accuracy of a model is often correlated with the ID accuracy.” citation needed


In Fig.1, does “ImageNet models” refer to both standard ImageNet classifiers and CLIP models trained on ImageNet? If so, isn’t it quite remarkable that CLIP models and standard classifiers lie on the exact same line? In general, it is unclear in this figure which of the models from section 5 are depicted here.


L128: Table 1 is a Figure not a Table


L181: “For datasets that consist of image-text pairs for language-image pre-training without original labels, including YFCC and LAION, we automatically generate classification labels by matching captions with ImageNet classes, and we then hold out a balanced test set from the original dataset.” Please provide examples of YFCC and LAION images+captions which were accepted into the ID test sets, along with their new ImageNet label in Appendix B.2.



#### Clarity:

Overall, the 3d plane plots are unhelpful in gauging whether points lie close to the plane or not. In addition to the provided 2d projections _along the ID accuracy axes_ a 2d projection orthogonal to the fitted plane could be helpful to visually judge the model fit.


L121: the sentence in line 121 is a bit hard to parse, would suggest rephrasing for clarity.


L133: the sentence in line 133 is a bit hard to parse, would suggest rephrasing for clarity.


L260-261: it is unclear that "SLIP models ... achieve higher average effective robustness" _compared to other models_, rather than _compared to single-ID setting_


Can the used OOD sets really be considered OOD sets for all training distributions? LAION, for example, contains numerous renditions and sketches. As a result, the measured effective robustness might not be a true “OOD robustness”. A remark on this terminology could be helpful.


#### Grammar / typos:

Abstract, typo: “Our new evaluation metric provides a better estimate of the effectiveness robustness”


L11: “effective~~ness~~ robustness”


L18: “following ~~on~~”


L22: “Unfortunately,” comma missing


L63: “are … resulted from” is grammatically incorrect


L78: “Limitation **of** the Single ID Test Set”



L80: “reflective **of**”



L81: “reflective **of**”


L81: “However, as the emergence of large-scale pre-trained models,” grammar, maybe “due to” instead of “as”?


L85: “when only one … **is** used”


L85: “--” should be “:”, I think


L88, 114, 117, 123, 172, 224, Table 3, Caption/L279: comma missing before respectively. Please add commas before all instances of “respectively”. In addition, if there is no full-stop after “respectively”, a comma should be added after this word as well (e.g., Table 3, caption).


L110: “test set”, set is missing


L121, L265: comma after thus missing


L148: “**generalized**”, “in **principle**”


L166: rephrase to “and subsample the combined datasets”


L195: should “retrain” be “retain”?


L258: “although **they** may”


L290: “models” should be plural


L315: may further be


L503: “**held** out”


L515: “**under**”


L519 “**clearer** understanding **of**”



**Questions:**


I am surprised how well models trained on CIFAR10 perform on ImageNet (x-axis in Figure 3b, blue dots). Was the size of the CIFAR10 images kept as usual or upscaled to 224x224 to make it more comparable to ImageNet trained models? I am surprised that training on 32x32 CIFAR10 images gives such high accuracy numbers on ImageNet, even after class-subsampling. Intuitively, I would expect the accuracy on CIFAR-10.2 to be much higher than on ImageNet, of a model trained on CIFAR10 since the distribution of CIFAR-10.2 should be closer to CIFAR10 than ImageNet. But the accuracies on CIFAR-10.2 and ImageNet are almost the same; the effect is much smaller than I would have expected. Could you maybe comment on this finding? Do you find it surprising / have any intuition or explanation for it?


Table 3 vs Table 2: In Table 2, the R² coefficients were all above 99% while in Table 3, most numbers are generally below 99%. Why do you think that is? For ObjectNet, the R² values for both YFCC and LAION are the lowest ones among all datasets. What makes ObjectNet different from the other datasets and why does the technique work worse for this dataset? Is there an interpretation of what a higher R² value means for the different training datasets, maybe in terms of similarity to the test set? E.g., on ObjectNet, the R² is higher when training on YFCC compared to LAION. Can anything interesting be deduced from this?


Following up on the previous point: Can there be a situation when your method does not work and could this be measured with the R² fitting score? Is it possible to pin it down to a threshold value when the technique should not be used? E.g. R² < 95% or R² < 90%?


Additionally, it is unclear to what extent the method relies on the ID sets to be similar. Do you have an intuition whether this approach would work at all for models trained on datasets that could be considered “OOD sets” of each other (e.g. a model trained on natural images vs. a model trained on sketches)? Related to this, how important was it for the fit of the baseline that models trained on a mixture of the ID sets were available? An ablation of this point would be interesting, as it might be prohibitively expensive in practice to retrain models on multiple mixtures of the ID sets simply to facilitate a comparison.


It seems to me as if the data would have been available to additionally facilitate a 3-way multi-ID evaluation of some models on the ID sets ImageNet, LAION, and YFCC. Could you elaborate why this was not attempted? I believe it would greatly strengthen the claim that this method should natively extend to more than 2 ID sets.


**Limitations:**

yes

---

> ### Author Rebuttal · Authors · 2023-08-10
>
> We want to thank the reviewer for their time and attention!
>
> We will fix the writing issues accordingly and make further proof-reading. We answer the questions below:
>
> ## CIFAR models on ImageNet
> When a model is trained on both CIFAR and ImageNet, we upsample CIFAR images to 224x224. If a model is only trained on CIFAR, 32x32 is used.
>
> The reviewer mentioned that "even after class-subsampling", but class-subsampling tends to increase the accuracy when there are fewer classes. When evaluating the class-subsampled ImageNet accuracy, we only evaluate on images with ImageNet labels that can be mapped to CIFAR-10 classes [26], which has potentially made this version of ImageNet much easier.
>
> ## $R^2$ values on different OOD datasets
> $R^2$ can be affected by the relationship between the ID datasets and the OOD datasets. It tends to be harder to predict the OOD performance when the OOD datasets have larger distribution shifts or are less related to the ID datasets and then $R^2$ values tend to be lower. As mentioned by the reviewer, it can have implications on the similarity between the datasets.
>
> A good fitting does not always exist. As mentioned in Line 273-275 and Line 308-311, even if the models are trained on the same dataset and the previous single-ID evaluation is used, "accuracy-on-the-line" does not always hold. Thus, we only consider datasets where at least "accuracy-on-the-line" holds for models trained on the same dataset. On these datasets, our method works well. As mentioned by the reviewer, it may be checked by $R^2$ scores and it is good to set a threshold. We think $R^2\geq 90\%$ is generally satisfactory, although one may set a different threshold if they have a different requirement.
>
> ## Similarity between the ID test sets
> We do not think our method is applicable when the training datasets are highly different. When we have multiple ID test sets, we are treating these test sets as ID test sets for all the models trained on different data. Thus, these ID test sets need to be relatively close compared to the OOD test sets. In our case, ImageNet and YFCC are relatively close, compared to OOD test sets that are quite different (e.g., ImageNet-Sketch). We will mention this restriction in the revision.
>
> ## Including models with mixed training data in the fitting
> We add an ablation study below. We evaluate the MAE scores of models trained on a mixture of datasets (ImageNet+YFCC models and ImageNet+LAION models), when they are excluded or included in the fitting respectively. The MAE scores (the lower the better) increase if the models with mixed training data are excluded in the fitting but are generally satisfactory. In this work, we have trained a large number of models with data mixed at various ratios and considered models with diverse combinations of accuracies, to obtain more convincing conclusions on the effective robustness.
>
> **MAE of ImageNet+YFCC models**
> | OOD test set | Excluding ImageNet+YFCC models in the fitting | Including ImageNet+YFCC models in the fitting |
> | --- | --- | --- |
> | ImageNet-V2 | 0.0071 | 0.0064 |
> | ImageNet-R | 0.0130 | 0.0121 |
> | ImageNet-Sketch | 0.0098 | 0.0094 |
> | ObjectNet | 0.0173 | 0.0095 |
>
> **MAE of ImageNet+LAION models**
> | OOD test set | Excluding ImageNet+LAION models in the fitting | Including ImageNet+LAION models in the fitting |
> | --- | --- | --- |
> | ImageNet-V2 | 0.0054 | 0.0051 |
> | ImageNet-R | 0.0156 | 0.0119 |
> | ImageNet-Sketch | 0.0262 | 0.0203 |
> | ObjectNet | 0.0291 | 0.0147 |
>
> ## 3-way multi-ID evaluation
> By comparing two datasets at a time in this work, we can already obtain conclusions that the considered models trained on any of the considered datasets have similar effective robustness, and a 3-way evaluation is not needed here. Directly comparing more than two datasets can be much more costly, as a larger number of models with diverse combinations of accuracies are needed for obtaining relatively convincing fitting planes and conclusions. We also mentioned it in Line 149-151 and Line 316-317.

---

> > ### Comment · Reviewer_MjfW · 2023-08-12
> > **Response to the rebuttal**
> >
> > Dear authors,
> >
> > thank you for responding to my concerns. My questions have been answered and I think this is a solid paper that should be accepted. Thus, I maintain my score of 7 and look forward to the reviewer discussion next week.
> >
> > Best,
> > Reviewer MjfW

---

> > > ### Author Response · Authors · 2023-08-15
> > > **Thanks to the reviewer**
> > >
> > > We thank the reviewer for responding to our rebuttal and supporting the paper!

---

### Official Review · Reviewer_sN9Q · 2023-07-06

**Soundness:** 3 good
**Presentation:** 3 good
**Contribution:** 3 good
**Rating:** 5
**Confidence:** 4

**Summary:**

This work delves into the definition of effective invariance, which is proposed to measure how the model performs beyond what we can expect from its in-distribution accuracy. This work points out that the training distribution should be considered when measuring the model's effective robustness. Specifically, models can be trained on different and diverse distributions. Then, the usage of the "in-distribution" test set should be changed accordingly. The new evaluation metric provides a more accurate characterization of CLIP models' robustness.

**Strengths:**

+ [***The motivation of re-define the effective invariance is sound***]  Large-scale models trained on various datasets make it necessary to evaluate and compare models trained on different data distributions. Namely, the in-distribution dataset should be well-aligned with the training distribution.

+ [***Extending the idea of effective invariance to multiple ID test sets is sound*** ] Using multiple ID test sets to check the relationship between ID accuracy and OOD accuracy is reasonable and straightforward.

+ [***Good clarify***] This work is well-written and easy to read. The motivation, background knowledge of effective invariance, and the idea of multiple ID test sets are well presented.

**Weaknesses:**

- [***Fitting quality is not very convincing***] In experiments, fit quality is mainly used to show that using multiple ID test sets can lead to a more accurate match between ID and OOD accuracy. If our goal is to produce more accurate fits, we can use one ID test set and one OOD test set to fit another OOD test set. Please comment on why you should care about the fitting quality.

- [***Discussing ID test sets for models pre-trained on more data***] The motivation for using multiple ID test sets is clear. However, for models pre-trained with more data and finetuned on ImageNet, what is their ID test set?

- [***Clarification of CIFAR model***] CIFAR images cost $32\times32$, while ImgeNet images are much higher resolution. How to combine them for training? Upsampling or downsampling?

- [***Please summarize observations***] It is unclear what the main observations and new insights are given by the multiple ID sets. It is clear that "the effective robustness of a zero-shot CLIP model trained on language-image data is similar to that of a standard ImageNet model". What other discoveries have been made besides this? For example, does pretraining on more data still contribute to effective invariance?

**Questions:**

Please comment on the choice of OOD test set:  CLIP models are trained on diverse and large-scale training sets (e.g., LIAON). Then, the test set may already be covered by the training distribution. Namely, a test set is OOD for ImageNet models but may be so OOD for CLIP models. In such a case, the effective invariance may truly reflect model behaviors under distribution shifts.

**Limitations:**

This work includes a paragraph to discuss the limitation and potential future directions, where the access to training distribution and determination of training distribution for some models are discussed. One more point to discuss: training distributions of CLIP models may already cover test distribution, then the definition of OOD needs to be re-considered.

---

> ### Author Rebuttal · Authors · 2023-08-10
>
> We want to thank the reviewer for their time and attention!
>
> ## Fitting quality
> A relatively high fitting quality is desired, which implies that the considered models have similar effective robustness and it is a major finding in this work.
>
> Following previous works on effective robustness, effective robustness checks the extra performance on OOD data beyond the performance that can be predicted from ID performance (Line 1~2). Therefore, there is a constraint here that the test sets used for the prediction cannot be arbitrarily taken, and they need to be ID test sets, not an OOD dataset. We only take ID test sets that match the training data, which is also consistent with the single-ID evaluation if models are only trained on a single dataset [40]. If an OOD test set such as ImageNet-R is used for the prediction, it already measures OOD performance. Though the fitting quality may be better, it cannot reveal the effective robustness.
>
> ## Models pre-trained on more data and fine-tuned on ImageNet
> For those models, we consider that their training distribution gradually shifts from the pre-training distribution to the ImageNet distribution, as the fine-tuning proceeds, and during the fine-tuning, the training distribution is some combination of the pre-training one and ImageNet. In this case, we use two ID test sets, including ImageNet and also the one matching the pre-training data, to cover the training distribution.
>
> ## Combining CIFAR and ImageNet in training
> We upsample CIFAR images to 224x224 for combined training. We will mention it in the paper.
>
> ## Main takeaways
> One main observation is the one already mentioned by the reviewer about the effective robustness of various models including zero-shot CLIP models. In addition, we propose that the effective robustness evaluation needs to be fully aware of the training data and consider the training data, when there are models trained on different datasets.
>
> The amount of training data does not affect effective robustness, as already mentioned in previous works [12].
>
> ## Training distribution of CLIP may overlap with the test distribution
> This is a limitation that also exists in previous works as long as they evaluate CLIP on the commonly adopted OOD datasets. Some previous works have mentioned that the likelihood of a direct data overlapping is low (e.g., see Section 6 in Schuhmann et al., 2022 or Section 5 in [30]). But it may require future work to better understand the effect. We will add it to the limitation section.
>
>
> Schuhmann, C., Beaumont, R., Vencu, R., Gordon, C., Wightman, R., Cherti, M., ... & Jitsev, J. (2022). Laion-5b: An open large-scale dataset for training next generation image-text models. Advances in Neural Information Processing Systems, 35, 25278-25294.
>
> Gadre, S. Y., Ilharco, G., Fang, A., Hayase, J., Smyrnis, G., Nguyen, T., ... & Schmidt, L. (2023). DataComp: In search of the next generation of multimodal datasets. arXiv preprint arXiv:2304.14108.

---

> > ### Comment · Reviewer_sN9Q · 2023-08-12
> > **Thanks for the rebuttal**
> >
> > Dear Authors,
> >
> > Thank you for providing the answer to my previous questions!
> >
> >  I was not asking about potential data overlap. Rather, distribution-level overlap. For example, CIFAR-10 training and test sets, they do have data overlap but their distribution is overlapped. Anyway, I appreciate you pointed out the discussion on data overlap in the other two works, which helps partially solve my initial concern.
> >
> > - Quick question: Based on the response, I am wondering: is it safe to say CLIP is not as robust as previous work claimed? Namely, there is still a long way to go in exploring models with highly effective robustness.
> >
> > After checking other reviewers' comments, I tend to my original rating of Borderline Accept and am willing to discuss it with other reviewers.
> >
> > Best,
> >
> > Reviewer sN9Q

---

> > > ### Author Response · Authors · 2023-08-15
> > > **Reply to the reviewer update**
> > >
> > > Dear Reviewer sN9Q,
> > >
> > > Thanks for your quick update and further elaborating the question! We agree that there could be a distribution-level overlap as you mentioned. As we mentioned in our initial rebuttal, this is an existing limitation of works using these OOD test sets for CLIP models, which will be interesting for future works to understand the effect. We show that even if there could be such a distribution-level overlap for CLIP models, CLIP models still do not have effective robustness gains.
> > >
> > > We only conclude that CLIP itself does not have real **effective robustness** gains. However, one may define robustness in other ways, e.g., one may consider the absolute accuracy on the existing OOD test sets as one of the metrics for robustness. CLIP, which enables pre-training on large-scale image-text data, still archives impressive absolute accuracy on these OOD test sets (however, this could be because of the distribution-level overlap as you mentioned). Therefore, our conclusion is more conservative and restricted to **effective robustness**.
> > >
> > > Thanks,
> > >
> > > Paper 4002 Authors

---

> > > > ### Comment · Reviewer_sN9Q · 2023-08-15
> > > > **Thank You**
> > > >
> > > > Dear Authors,
> > > >
> > > > Thanks for the clarification and insights. I do not have any other concerns.
> > > >
> > > > Best,
> > > >
> > > > Reviewer sN9Q

---

### Official Review · Reviewer_Kcyx · 2023-07-07

**Soundness:** 2 fair
**Presentation:** 3 good
**Contribution:** 3 good
**Rating:** 6
**Confidence:** 3

**Summary:**

The authors identify that the definition of effective robustness is ill-equipped for comparing models trained on a variety of distributions; in particular, it only uses a single data distribution to predict OOD accuracy $acc_{OOD}$. The authors propose a more precise effective robustness measurement for models trained on two different datasets ($data_1^{train}$ and $data_2^{train}$), which is based on using regression to optimize the parameters $b,w_1, w_2$ in the following equation: $acc_{OOD}(model_i) = b + w_1 acc_{data_1^{test}}(model_i) + w_2 acc_{data_2^{test}}(model_i),$ where accuracies are given in logit space. Thus, if we wish to compare the effective robustnesses of models $A$ and $B$ that were trained on $data_1^{train}$ and $data_2^{train}$, respectively, we can use this new approach to predicting $acc_{OOD}$ values without being forced to use just one of the two data distributions ($data_1$ or $data_2$) as the distribution we use to predict $acc_{OOD}$. The benefit of this approach, across a wide variety of contexts/models, is that it fits the data better and gives lower effective robustness measurements for models that appeared effectively robust under the old definition because they were trained on a dataset from a distribution different from the one used to predict $acc_{OOD}$.

**Strengths:**

Originality:

1. The submission's effective robustness (ER) definition is a simple and novel extension of the original definition.

Quality:

2. The first three sections of the paper are strong. In particular, Section 3 is a very convincing and interesting illustration of a problem with our current definition of effective robustness.

3. The submission's approach (described in Section 4) provides interesting results that may be helpful for understanding when a model is effectively robust. The models and datasets used are comprehensive.

Clarity:

4. Largely, the paper is well written and organized.

Significance

5. This work offers an important insight about evaluation protocol in an active area of research.

**Weaknesses:**

My main concern is that, despite superficial similarity in their definitions, the interpretations of single- and multi-ID ER are drastically different. Accordingly, I am concerned about whether the new ER definition (multi-ID ER) will have practical utility for researchers interested in identifying models with ER. I expand on these concerns below and in the "Questions" section.

1. My (and seemingly the authors') understanding of the purpose of effective robustness is that it identifies models that surpass the OOD accuracy we expect of them *based on their ID accuracy* (i.e., we want to know if training on ID data using some new technique produces more robustness than expected). In Section 3, this submission clarifies that ER does not serve this purpose when we fix the "ID" dataset in our definition of ER to be $data_1^{test}$, then use that ER to compare models trained on ID data ($data_1^{train}$) with models trained on OOD data ($data_2^{train}$).  However, this submission's solution also seems to lose this original purpose: models with high multi-ID ER are those with higher-than-expected $acc_{OOD}$ when predictions are made using a combination of an **OOD accuracy** (for a model trained on $data_1^{train}$, this would be $acc_{data_2^{test}}$) and an ID accuracy (for a model trained on $data_1^{train}$, this would be $acc_{data_1^{test}}$). In other words, this metric's use of an OOD accuracy (e.g., $acc_{data_2^{test}}$ for the model trained on $data_1^{train}$) to predict an OOD accuracy (i.e., $acc_{OOD}$) seems to preclude its interpretation as an ER measurement and harm usefulness/interpretability. For example, because we would expect OOD accuracies to be correlated, it seems that the calculation of multi-ID ER can lower the detected ER in a model that truly has ER: if a model has high OOD accuracy (e.g., $acc_{data_2^{test}}$ for a model trained on $data_1^{train}$), that can inflate multi-ID ER's prediction of $acc_{OOD}$, which lowers the detected ER of the model.

**Questions:**

1. Many of the results are intuitive/interesting, but I'm not sure how they are aligned with the notion of predicting OOD performance from an ID dataset (i.e., "effective robustness"). My understanding of the proposed definition of multi-ID effective robustness is that it involves predicting OOD performance using an ID *and* an OOD performance, rather than using two ID performances as the proposed name ("multi-ID" ER) suggests. I think the submission (and my score) could be significantly improved if the authors can clarify why multi-ID ER can be interpreted as an ER measurement without issues.

2. While the original ER definition is limited in ways (including a way identified by this submission), it also has clear practical utility in that I can understand whether a model outperforms on OOD data relative to models trained on the same ID data. The practical utility of this submission's proposed alternative ER definition is unclear to me. For instance, given a model trained on $data_1$, is it possible to get a different understanding of its multi-ID ER simply by changing the second dataset used in the ER definition (e.g., from YFCC to LAION)? If so, how many dataset pairs ($data_1$, $data_i$) do I need to check my model on before I understand its expected OOD performance (or its ER)?

3. Line 264: Why would WiSE-FT alter the underlying training distribution in a way that makes it unsuitable for multi-ID ER? It seems like its train distribution would be similar to that of the models that are trained on dataset mixtures.

4. Why not report the R2 in some tables (e.g., Table 5)?

5. Table 1 should be labeled as a Figure.

6. It might be nice to see the plane equations on the plane plots.

**Limitations:**

Potential limitations are well addressed.

---

> ### Author Rebuttal · Authors · 2023-08-10
>
> We want to thank the reviewer for their time and attention!
>
> ## Two ID test sets
> In this paper, we consider that two ID test sets are used, rather than an ID test set and an OOD test set as mentioned by the reviewer. For models trained on one of the ID datasets, even though there may be some distribution shift between their training data and the other ID test set, we assume that this distribution shift is relatively minor compared to the distribution shift to the OOD test sets (e.g., the difference between ImageNet and YFCC is much smaller, compared to the difference between ImageNet-Sketch and ImageNet/YFCC). Indeed this treatment is not perfect and can have limitations as the (potentially minor) distribution shift between the multiple ID test sets is not considered, and we will add it to the "limitations" section. But we believe it is still a better evaluation than the single-ID evaluation when models are trained on different data.
>
> ## Choosing the second dataset
> The choice of the second ID test set is not arbitrary. In this paper, we propose that the ID test sets need to match the training data (Line 113~114). So if we are comparing ImageNet and YFCC models, the ID test sets will only be from ImageNet and YFCC, but not LAION. When we only choose ID test sets in this way, we show that it is already enough to conclude that the considered models mainly have similar effective robustness.
> Choosing the ID test sets in some different way (e.g., as mentioned by the reviewer) will be a different setting. It is still interesting for future work to consider, but it is beyond the scope of our proposed method.
>
> ## WiSE-FT
> WiSE-FT performs weight-space ensembling which edits the weights. It is not quite clear what the training distribution is for such kinds of models with post-training weight editing, compared to models that are directly trained from a known distribution without weight modification (e.g., dataset mixture).
>
> ## $R^2$ in Table 5
> $R^2$ values are not reported as the models in the table are not involved in the fitting. They are directly evaluated on fitting lines/planes from Table 3.
>
> ## Figures
> We will change Table 1 into Figure 1, and add plane equations to the figures.

---

> > ### Comment · Reviewer_Kcyx · 2023-08-16
> > **Acknowledgement of rebuttals and reviews**
> >
> > I have read the reviews and rebuttals, and I maintain my original score. Based on the rebuttal, I have some new comments/questions, and an idea for an experiment that might clarify the benefits of the submission's methodology. Presently, I am inclined to find a reason to advocate for this paper because (as I said in my review) its Section 3 provides a convincing and novel insight into an important problem (ER comparisons among models with different training data).
> >
> > >we assume that this distribution shift is relatively minor compared to the distribution shift to the OOD test sets
> >
> > A small distribution shift (like the one between YFCC and ImageNet) that degrades performance by a small amount for some models and by an even smaller amount for other models indicates that the latter set of models is more robust to OOD shifts. Thus, the size of the distribution shift relative to other potential shifts is irrelevant to my critique: in other words, a small distribution shift can indicate the presence of OOD robustness as well as a large distribution shift can.
> >
> > In sum, I disagree with the idea that YFCC is ID for ImageNet models and think the rebuttal's proposal to address small differences in these distributions with a statement in the paper's limitations section is insufficient. I would recommend not arguing that YFCC is ID for ImageNet models, renaming “multi-ID” ER accordingly (e.g., "multi-D" ER?), and addressing my remaining concern, which follows next.
> >
> > >we believe it is still a better evaluation than the single-ID evaluation when models are trained on different data.
> >
> > The idea that the proposed multi-ID evaluation is better than single-ID evaluation when models are trained on different data requires explanation/justification beyond what was given in the paper. To me, there’s a tradeoff, and it remains unclear which method is better. Multi-ID evaluation seems biased towards deflating ER (for the reasons I gave in my review), while single-ID evaluation seems biased towards inflating ER (see the submission's Section 3).
> >
> > To clarify, I think the purpose of ER is to aid the identification of models (or, more broadly, training configurations) that improve robustness. Is there a way to show that multi-ID ER does not hinder my ability to find such models? For example, some ImageNet models will have relatively high **single-ID** ER when compared to other ImageNet models – can you ensure that these same models have relatively high **multi-ID** ER when compared to other ImageNet models? Showing that models' relative ER rankings are consistent in a statistically-significant way (e.g., with a Kendall rank coefficient test) would address my primary concern (the interpretability of multi-ID ER as an ER). If this makes sense, I would want to see this evidence for the YFCC and LAION models too. I am also happy to chat about your thoughts on this potential experiment and/or other ways the submission might improve along these lines (if you have questions/suggestions).
> >
> > >Choosing the ID test sets in some different way (e.g., as mentioned by the reviewer) will be a different setting.
> >
> > This is minor compared to the above concerns, but I am not suggesting choosing the second ID test set in a different way. I am just pointing out that there are many possible options for a second ID test set, and all of them will potentially give you a different understanding of the ERs of the models trained on the first ID test set (the experiment I suggested in the prior paragraph will help address this point, which is ultimately about the practicality of multi-ID ER).

---

> > > ### Author Response · Authors · 2023-08-21
> > > **Response to the additional questions**
> > >
> > > Thanks to the reviewer for the additional feedback.
> > >
> > > The reviewer has asked whether some ImageNet models with relatively high single-ID ER will still have relatively high multi-ID ER. We want to clarify a misunderstanding here. Previous works have concluded that existing models trained on the same dataset have similar single-ID effective robustness [40, 12, 29]. Therefore, "some ImageNet models will have relatively high single-ID ER when compared to other ImageNet models" mentioned by the reviewer does not really hold. It is the same for YFCC models or LAION models [12, 29].
> > >
> > > Though there can be some distribution difference between the multiple ID test sets we use, we are assuming that this difference is sufficiently small, compared to the distribution shift to the OOD test sets. If there exists some model with noticeably higher effective robustness, such a model tends to have stronger relative robustness gains when the distribution shift is larger, compared to other less robust models. Therefore, we believe that the size of the distribution shift still matters, and as long as the potential distribution shift between the ID test sets is small, we expect the effect of "deflating ER" to be very small.
> > >
> > > If a model with an effective robustness gain does exist, such a model is still expected to achieve higher OOD test accuracy (on a OOD test set with a relatively large distribution shift) compared to what is predicted by the accuracy on the two ID test sets (the ID test sets may at most have a relatively small distribution shift). Thus we believe the new effective robustness does not hinder the ability to find effectively more robust models.
> > >
> > > Finally, we agree that the "multi-ID effective robustness" term can be revised into a more precise one, potentially "training data-aware effective robustness".

---

> > > > ### Comment · Reviewer_Kcyx · 2023-08-21
> > > > **No quantitative evidence to support claims**
> > > >
> > > > > We want to clarify a misunderstanding here. Previous works have concluded that existing models trained on the same dataset have similar single-ID effective robustness [40, 12, 29].
> > > >
> > > > In Figure 2 of [40], the authors show that models trained on ImageNet with standard techniques can have the same ID accuracy while displaying 5+ percentage point differences in performance on ImageNet-Vid-Robust-anchor data and 10+ percentage point differences in performance on YTBB-Robust-anchor data. The authors of [40] say: “The linear fit is best for ImageNetV2, ObjectNet, and ImageNet-Vid-Robust with respective r2 scores of 1.00, 0.95, and 0.95.” In other words, even when the fit is strong at 0.95 r2, there can be substantial variation (5% or more) in single-ID effective robustness.
> > > >
> > > > Given these large differences, I still support my original statement that some models have relatively high single-ID ER, while the authors suggest that this statement “does not really hold”:
> > > > > Therefore, "some ImageNet models will have relatively high single-ID ER when compared to other ImageNet models" mentioned by the reviewer does not really hold
> > > >
> > > > My original rating (5) was based on my assumption that my concerns were possibly exaggerated and would be easily addressed by the authors. The author rebuttal and discussion made a case for the submission via unsupported claims, which in some cases are demonstrably false (see my comments on [40] above). Overall, the author responses have left me unable to continue giving the manuscript's methodology the benefit of the doubt, and I have updated my score accordingly. Notably, my concerns were for the last 6 days and still are able to be addressed via easily obtainable quantitative evidence that can be generated without training new models, using a simple statistical test on existing data.

---

> > > > > ### Author Response · Authors · 2023-08-21
> > > > > **Our response is based on the conclusions from [40]**
> > > > >
> > > > > Our response was based on the fact that [40] has not recognized that the models mentioned by the reviewer are effectively more robust. [40] says:
> > > > >
> > > > > >Moreover, most current techniques provide no robustness to the natural distribution shifts in our testbed. The main exception is training on larger and more diverse datasets, which in multiple cases increases robustness, but is still far from closing the performance gaps.
> > > > >
> > > > > That is, [40] concluded that only changing the training data can improve the single-ID effective robustness in multiple cases. At least, the ImageNet models mentioned by the reviewer do not reliably exhibit consistent robustness gains across various datasets, especially on the datasets we are adopting in this work.
> > > > >
> > > > > ImageNet-Vid-Robust-anchor and YTBB-Robust-anchor mentioned by the reviewer are not included in this work. We have mentioned in the paper that we focus on OOD datasets where "accuracy-on-the-line" from the single-ID effective robustness holds well for models trained on the same data (Line 308-310), i.e., models trained on the same data have similar effective robustness. And for these cases, we aim to improve the effective robustness evaluation when there are models trained on different data.

---

> > > > > > ### Comment · Reviewer_Kcyx · 2023-08-21
> > > > > >
> > > > > > Let's discuss this in terms of the red-colored models plotted in the submission's Figure 3b. Note that I am using approximate numbers because I am guessing at values based on the figure. Also, there are many models plotted in Figure 3b, but models A and B in the following table seem to represent two of them.
> > > > > >
> > > > > > | Model | ID Dataset | OOD Dataset | ID Accuracy | OOD Accuracy | Single-ID ER | Multi-ID ER
> > > > > > | --- | ----------- | ----------- |  ----------- |  ----------- |  ----------- |  ----------- |
> > > > > > | A | ImageNet | CIFAR-10.2 |  ~60% |  ~50% |  +3% |  ??
> > > > > > | B | ImageNet | CIFAR-10.2 | ~60% |  ~44% |  -3% |  ??
> > > > > >
> > > > > > Clearly, A has more ER than B. A will probably have more ER than a lot of models: its ER is relatively high. If you fill in this table with the multi-ID ER values, do you see that A has higher multi-ID ER than B? In other words, are relative ER rankings preserved? If this makes sense, I would perform this comparison using all models (not just A and B) in a more rigorous way (e.g. Kendall's rank correlation test), then repeat this experiment for the other OOD dataset and ID dataset combinations.
> > > > > >
> > > > > > Such experiments are easy to run on existing data and would address my review's main concern. More importantly, they would quantitatively tell readers the extent to which multi-ID ER aligns with single-ID ER, which would help readers interpret the behavior of the former.

---

> > > > > > > ### Author Response · Authors · 2023-08-22
> > > > > > > **Results on Kendall's rank correlation test**
> > > > > > >
> > > > > > > We thank the reviewer for the specific suggestions. We have conducted a Kendall's rank correlation test to compare single-ID effective robustness and multi-ID effective robustness. We report the $\tau$ statistic (computed by `scipy.stats.kendalltau`):
> > > > > > >
> > > > > > > CIFAR-10 v.s. ImageNet
> > > > > > > | | CIFAR-10.1 | CIFAR-10.2 | CINIC-10 |
> > > > > > > | --- | --- | --- | --- |
> > > > > > > | CIFAR-10 models | 0.9619 | 0.6667 | 0.8095 |
> > > > > > > | ImageNet models | 0.7496 | 0.5789 | 0.0640 | |
> > > > > > > | CIFAR-10 + ImageNet models | 0.6823 | 0.2599 | 0.6195 |
> > > > > > >
> > > > > > > ImageNet v.s. YFCC
> > > > > > > | | ImageNet-V2 | ImageNet-R| ImageNet-Sketch | ObjectNet |
> > > > > > > | --- | --- | --- | --- | --- |
> > > > > > > | ImageNet models | 0.5142 | 0.4412 | 0.8031 | 0.8063 |
> > > > > > > | YFCC models | 0.2307 | 0.4358 | 0.6153 | 0.5897 |
> > > > > > > | ImageNet + YFCC models | 0.4591 | 0.3305 | 0.5555 | 0.2438|
> > > > > > >
> > > > > > > ImageNet v.s. LAION
> > > > > > > | | ImageNet-V2 | ImageNet-R| ImageNet-Sketch | ObjectNet |
> > > > > > > | --- | --- | --- | --- | --- |
> > > > > > > | ImageNet models | 0.3123 | 0.4384 | 0.6216 | 0.9729 |
> > > > > > > | LAION models | 0.6483 | -0.5384 | -0.7802 | 0.6043 |
> > > > > > > | ImageNet + LAION models | 0.3246 | -0.1610 | -0.0129 | 0.4389 |
> > > > > > >
> > > > > > > Results suggest that the rankings on the single-ID effective robustness and multi-ID effective robustness are indeed positively correlated for CIFAR-10 and ImageNet models.
> > > > > > >
> > > > > > > Some of the $\tau$ statistics for models trained with LAION are negative, which is also reasonable, as the single-ID effective robustness using ImageNet as the ID test set is not a good evaluation for LAION models.

---

> > > > > > > > ### Comment · Reviewer_sN9Q · 2023-08-22
> > > > > > > >
> > > > > > > > Dear Authors:
> > > > > > > >
> > > > > > > > Could you please discuss the original question raised by reviewer Kcyx in light of your correlation results? I think this might help address Reviewer Kcyx's concerns. ImageNet models may all follow a general trend: their ER is not high, but there may be some models with specific schemes (such as data augmentation or special training scheduler) that are somewhat derived from this trend. Using multiple IDs can “bury” these models and make users overlook some underlying techniques.
> > > > > > > >
> > > > > > > > I attached the original comment from Reviewer Kcyx:
> > > > > > > >
> > > > > > > > >To clarify, I think the purpose of ER is to aid the identification of models (or, more broadly, training configurations) that improve robustness. Is there a way to show that multi-ID ER does not hinder my ability to find such models? For example, some ImageNet models will have relatively high single-ID ER when compared to other ImageNet models – can you ensure that these same models have relatively high multi-ID ER when compared to other ImageNet models? Showing that models' relative ER rankings are consistent in a statistically-significant way (e.g., with a Kendall rank coefficient test) would address my primary concern (the interpretability of multi-ID ER as an ER). If this makes sense, I would want to see this evidence for the YFCC and LAION models too. I am also happy to chat about your thoughts on this potential experiment and/or other ways the submission might improve along these lines (if you have questions/suggestions).
> > > > > > > >
> > > > > > > >
> > > > > > > > Moreover, I have read your comment: *"Though there can be some distribution difference between the multiple ID test sets we use, we are assuming that this difference is sufficiently small, compared to the distribution shift to the OOD test sets. If there exists some model with noticeably higher effective robustness, such a model tends to have stronger relative robustness gains when the distribution shift is larger, compared to other less robust models. Therefore, we believe that the size of the distribution shift still matters, and as long as the potential distribution shift between the ID test sets is small, we expect the effect of "deflating ER" to be very small"*
> > > > > > > >
> > > > > > > > It sounds like the choice of ID test set might be a potential limitation: For example, one set of models is trained on ImageNet-R and the other on ImageNet-S, when testing them on other datasets, how do we use multi-ID ER?
> > > > > > > >
> > > > > > > > Best,
> > > > > > > >
> > > > > > > > Reviewer sN9Q

---

> > > > > > > > > ### Author Response · Authors · 2023-08-22
> > > > > > > > > **Answering Reviewer sN9Q**
> > > > > > > > >
> > > > > > > > > To Reviewer sN9Q:
> > > > > > > > >
> > > > > > > > > We hope our additional results on the Kendall's rank correlation test could address the concerns in Reviewer Kcyx's original question. The results are showing that there is generally a good correlation between the relative ER rankings under the single-ID effective robustness and the multi-ID effective robustness for CIFAR-10/ImageNet/YFCC models. That is, models that have slightly more ER under the single-ID evaluation also tend to have slightly higher ER under the multi-ID evaluation. For LAION models, sometimes the correlation could be negative, but it is potentially because the single-ID evaluation is sometimes poor for LAION models even if we use the LAION ID test set for the single-ID evaluation.
> > > > > > > > >
> > > > > > > > > > It sounds like the choice of ID test set might be a potential limitation: For example, one set of models is trained on ImageNet-R and the other on ImageNet-S, when testing them on other datasets, how do we use multi-ID ER?
> > > > > > > > >
> > > > > > > > > ImageNet-R and ImageNet-Sketch are only used as OOD test sets as they are quite different from regular training data (ImageNet, YFCC, LAION) due to renditions in ImageNet-R and sketches in ImageNet-Sketch. Note that we are assuming that models are only trained on relatively normal data (such as ImageNet, YFCC, LAION, which basically contain normally crawled images from social media), and then the training distributions of all the models are relatively close compared to the OOD test sets. We will revise the paper to make this point more clear.

---

> > > > > > > > > > ### Comment · Reviewer_sN9Q · 2023-08-22
> > > > > > > > > >
> > > > > > > > > > Dear Authors,
> > > > > > > > > >
> > > > > > > > > > Thanks for the reply. I noticed Reviewer Kcyx already said the correlation result helps.
> > > > > > > > > >
> > > > > > > > > > As for using ImageNet-S/R as the training sets, I was giving you a specific example to illustrate two training sets could be significantly different. In this case, the proposed multi-ID ER would meet some limitations. That said, please clearly highlight that "the training distributions of all the models are relatively close compared to the OOD test sets" in the revised version.
> > > > > > > > > >
> > > > > > > > > > Kind Regards,
> > > > > > > > > >
> > > > > > > > > > Reviewer sN9Q

---

> > > > > > > > ### Comment · Reviewer_Kcyx · 2023-08-22
> > > > > > > > **This helps**
> > > > > > > >
> > > > > > > > Thank you for adding these new results! I think they show that, typically, the phenomenon that I was concerned about does not occur.
> > > > > > > >
> > > > > > > > >Some of the statistics for models trained with LAION are negative, which is also reasonable, as the single-ID effective robustness using ImageNet as the ID test set is not a good evaluation for LAION models.
> > > > > > > >
> > > > > > > > I agree with the idea that a negative rank correlation can be discounted if it is based on a relationship with a bad fit (i.e., a low r2 value). However, to clarify, the LAION models' single-ID ER numbers in this experiment should be computed using the LAION dataset ID accuracies (see my updated table below). The idea is to compare a non-controversial ranking of LAION models---single-ID ER with the ID dataset coming from the distribution of the model's training dataset---to a novel ranking of LAION models (i.e., multi-ID ER). Perhaps this is what you did, and I misunderstood your comment about using "ImageNet as the ID test set... for LAION models".
> > > > > > > >
> > > > > > > > Relatedly, in the "CIFAR-10 v.s. ImageNet" table, is the ImageNet row using ImageNet models' CIFAR-10 accuracies to compute single-ID ER? If so, this row would have the same issue as the aforementioned LAION row. To have your ImageNet row in your first table correspond to the experiment I suggested, let X1=ImageNet and let X2=CIFAR-10 in the following table:
> > > > > > > >
> > > > > > > > | Model | ID Dataset | OOD Dataset | ID Accuracy | OOD Accuracy | Single-ID ER according to regression model trained on dataset X1 accuracies | Multi-ID ER using model trained on accuracies from dataset X1 and dataset X2
> > > > > > > > | --- | ----------- | ----------- |  ----------- |  ----------- |  ----------- |  ----------- |
> > > > > > > > | A | X1 | ABC |  ~60% |  ~50% |  +3% |  ??
> > > > > > > > | B | X1 | ABC | ~60% |  ~44% |  -3% |  ??

---

> > > > > > > > > ### Author Response · Authors · 2023-08-22
> > > > > > > > > **Additional results for Reviewer Kcyx**
> > > > > > > > >
> > > > > > > > > In our previous results, we were only using the first ID test (i.e., CIFAR-10 when comparing CIFAR-10 and ImageNet, ImageNet when comparing ImageNet and YFCC/LAION), as this is the setting used in the original single-ID evaluation. We now add results when we use the second ID test set, as suggested by the reviewer:
> > > > > > > > >
> > > > > > > > > CIFAR-10 v.s. ImageNet (ImageNet as the ID test set for single-ID evaluation)
> > > > > > > > > |  | CIFAR-10.1 | CIFAR-10.2 | CINIC-10 |
> > > > > > > > > | --- | --- | --- | --- |
> > > > > > > > > | single-ID $R^2$ | 0.5802 | 0.6632 | 0.6509 |
> > > > > > > > > | $\tau$ for ImageNet models | 0.1664 | 0.1522 | 0.4907 |
> > > > > > > > >
> > > > > > > > > ImageNet v.s. YFCC (YFCC as the ID test set for single-ID evaluation)
> > > > > > > > > | | ImageNet-V2 | ImageNet-R| ImageNet-Sketch | ObjectNet |
> > > > > > > > > | --- | --- | --- | --- | --- |
> > > > > > > > > | single-ID $R^2$ | 0.9801 | 0.9639 | 0.9266 | 0.9178 |
> > > > > > > > > | $\tau$ for YFCC models | 0.0256 | 0.8461 | 0.7179 | 0.2564 |
> > > > > > > > >
> > > > > > > > > ImageNet v.s. LAION (LAION as the ID test set for single-ID evaluation)
> > > > > > > > > | | ImageNet-V2 | ImageNet-R| ImageNet-Sketch | ObjectNet |
> > > > > > > > > | --- | --- | --- | --- | --- |
> > > > > > > > > | single-ID $R^2$ | 0.6846 | 0.9714 | 0.8999 | 0.5115 |
> > > > > > > > > | $\tau$ for LAION models | -0.4945 | 0.6483 | -0.2747 | 0.6483 |
> > > > > > > > >
> > > > > > > > > We notice that when we change the ID test set here, $R^2$ values for the single-ID evaluation often become lower.

---

> > > > > > > > > > ### Comment · Reviewer_Kcyx · 2023-08-22
> > > > > > > > > > **Grateful for the requested results and clarity**
> > > > > > > > > >
> > > > > > > > > > Please incorporate something like the following block of quoted text in the limitations section of the manuscript (and include the experimental results in the appendix). I would be comfortable recommending acceptance for this paper given the addition of such caveats and text changes to weaken the claims*. Alternatively, if the authors do not agree with the following quoted text block, please let me know how their interpretation differs.
> > > > > > > > > >
> > > > > > > > > > >Often (for most models and datasets), we find that multi-ID ER is high when single-ID ER is high. In such cases, we view multi-ID ER as preferable to single-ID ER because it reflects model ER while allowing for comparisons across models trained on different datasets. However, a limitation of our methodology is that, for some dataset and model combinations, multi-ID ER does not reliably reflect single-ID ER. In these cases, ER properties of models cannot be reliably inferred from multi-ID ER alone. Therefore, we recommend using multi-ID ER only as a complement to single-ID ER, and we emphasize that relying on multi-ID ER alone will sometimes lead to counterintuitive conclusions (see examples below).
> > > > > > > > > > >
> > > > > > > > > > >1. Assume you are comparing a set of models that were trained on either LAION or ImageNet. We found that higher multi-ID ER LAION models will tend to have lower ImageNet-Sketch ER.
> > > > > > > > > > >
> > > > > > > > > > >2. Assume you are comparing a set of models that were trained on either YFCC or ImageNet. We found that YFCC models with higher multi-ID ER are about as likely to have lower ER as they are to have higher ER for ImageNet-V2.
> > > > > > > > > >
> > > > > > > > > > *I encourage the authors to seriously revise the statements they make in their manuscript to reflect the fact that multi-ID ER is not necessarily associated with ER. For example, the following statement in the abstract should be revised from:
> > > > > > > > > >
> > > > > > > > > > >Our new evaluation metric provides a better estimate of the effectiveness robustness and explains the surprising effective robustness gains of zero-shot CLIP-like models exhibited when considering only one ID dataset
> > > > > > > > > >
> > > > > > > > > > to something like:
> > > > > > > > > >
> > > > > > > > > > >Our new evaluation metric provides a different estimate of the effectiveness robustness and may explain the surprising effective robustness gains exhibited by zero-shot CLIP-like models when considering only one ID dataset.

---

> > > > > > > > > > > ### Author Response · Authors · 2023-08-22
> > > > > > > > > > > **Thanks for the suggestions and we will revise the paper accordingly**
> > > > > > > > > > >
> > > > > > > > > > > We thank the reviewer again for the suggestion. We agree that our multi-ID evaluation is not meant to replace the single-ID evaluation. Our new evaluation is basically proposed for comparing models trained on different datasets. To compare models trained on the same dataset, we would still recommend users to mainly use the single-ID evaluation, and the multi-ID results may be used as an optional and supplementary reference. To obtain a comprehensive analysis, we would recommend using both single-ID evaluation and multi-ID evaluation. As discussed here and suggested by the reviewer, we will revise the paper to include the results and explanations discussed here, state the restrictions and applicability of the multi-ID evaluation more clearly, and also reflect the restrictions in our statements and claims. We thank the reviewer for deeply engaging in the discussion and would appreciate it if the reviewer could update the rating accordingly.

---

### Official Review · Reviewer_mbLQ · 2023-07-26

**Soundness:** 2 fair
**Presentation:** 2 fair
**Contribution:** 2 fair
**Rating:** 5
**Confidence:** 3

**Summary:**

This paper extends the work "Measuring Robustness to Natural Distribution Shifts in Image Classification" to more than one ID test set scenario. This paper proposes a robustness evaluation metric that uses more ID test sets covering the evaluated models' training distributions.

**Strengths:**

1. The problem of Effective Robustness is important.

2. The paper's findings are interesting: vision-language model evaluation needs future explored given the unique training setting.

**Weaknesses:**

1 **The motivation is not sound**. Intuitively, the effective robustness evaluation is ID training dataset driven. If we want to compare different model architectures, comparing the domain shift OOD performance of different models trained on the same training dataset is meaningful. In that case, using a single ID test set is enough.

For models trained on large-scale image-language pair datasets (CLIP pretrained on LAION), it is unclear how to define a domain-shift OOD evaluation: how to find a "domain shift" OOD dataset of LAION? If the model is pertained on LAION and then finetuned on ImageNet, how to guarantee ImageNet-v2 is not overlapped with LAION? The accuracy of ImageNet-v2 itself is meaningful. Comparing different vision-language models that both trained on LAION is also meaningful. However, it is hard to understand why we must compare two models trained on different training sets. Which expected conclusion are you trying to get? The conflicting findings and conclusion on L95~L108 do not make sense if the previous motivation is unclear.

Usually, if we want to see the OOD performance of vision-language models, we show a list of performances on different test sets.

2 **The Figure-1 and conclusion are confusing** In Figure1, how many models are you using as ImageNet models and YFCC models? Why do you want to compare these two groups of models? What is your expected possibility of conclusion?
Also, in Figure1(a), the ImageNet accuracy overlap between the two groups is limited, so it is hard to directly compare the two fitted lines.

3 **Limited Novelty** If we skip the motivation problem and focus on the proposed method, it is straightforward to extend equation 1 to equation 3 if we want to consider two ID test sets.

4 **Generalization to more than two ID tests** The paper only conducted experiments on two ID test sets, but claimed the effectiveness on multiple ID test sets. It is not straightforward to extend to more datasets. For instance, how to build the sampling classes in Section 5.2? How to guarantee the sampled overlap class is good enough to evaluate

**Questions:**

See above

---

> ### Author Rebuttal · Authors · 2023-08-10
>
> We want to thank the reviewer for their time and attention!
>
> ## Why do we compare the effective robustness of models trained on different datasets?
> At a general level, state-of-the-art models are increasingly being trained on large swaths of private data sources, but despite these differences in training distributions their performance is regularly compared, often in leaderboards composed of challenging benchmarks.  Because the field is heading in this direction, we believe it is critical to understand what conclusions can and can’t be drawn from such cross-model and training-data-agnostic evaluations.
>
>
> With respect to robustness, this issue has already arisen in prior work.  As mentioned in Line 29-36, previously the well-known CLIP paper [30] has claimed unprecedented effective robustness gains of CLIP models, and [12, 29] later studied the reasons behind the effective robustness gains of CLIP models. These works all compared the effective robustness of ImageNet models and CLIP models pre-trained on other data (under the single-ID evaluation), and [12, 29] concluded that training data affect the effective robustness.
>
> Given these previous works, the motivation here is that we aim to further understand if it is truly enough to compare the effective robustness of models trained on different data using the previous single-ID evaluation, and if changing the training data can truly improve effective robustness. We have found the limitations of the previous evaluation and proposed a new evaluation. This work is not raising the problem of comparing the effective robustness of models trained on different data for the first time, but it is providing a new understanding on the existing problem that has been studied by multiple existing works [30, 12, 29]. We conclude that the evaluation needs to be aware of the training data, and changing the training dataset does not improve the effective robustness. Therefore, we believe the motivation of this work is well supported given the previous works [30, 12, 29].
>
> ## Which datasets are really OOD?
> While we agree with the reviewer that there is a risk of overlap between different datasets, we believe this a common challenge in defining “out-of-distribution” and that the datasets we chose both follow best practice and still shed meaningful conclusions.
>
> We have used several OOD test sets in this work, such as ImageNet-R, ImageNet-Sketch, etc. It is also consistent with many previous works studying LAION models, including the [29], the LAION-5B paper (Schuhmann et al., 2022), Gadre et al., 2023, etc.  Further, previous works have already found that the likelihood of data overlapping is low (e.g., see Section 6 in Schuhmann et al., 2022 or Section 5 in CLIP [30]).
>
> Schuhmann, C., Beaumont, R., Vencu, R., Gordon, C., Wightman, R., Cherti, M., ... & Jitsev, J. (2022). Laion-5b: An open large-scale dataset for training next generation image-text models. Advances in Neural Information Processing Systems, 35, 25278-25294.
>
> Gadre, S. Y., Ilharco, G., Fang, A., Hayase, J., Smyrnis, G., Nguyen, T., ... & Schmidt, L. (2023). DataComp: In search of the next generation of multimodal datasets. arXiv preprint arXiv:2304.14108.
>
> ## Figure 1
> These models are also used in Table 4 which has mentioned that there are 36 ImageNet models and 13 YFCC models. We will also add this information in Figure 1.
>
> "Why do you want to compare these two groups of models?" It follows [12] which also compared ImageNet models and YFCC models. YFCC is a public dataset with image-text pairs and thus it enables CLIP training which is previously believed to improve effective robustness under the single-ID evaluation.
>
> "What is your expected possibility of conclusion?" When the ID test set is changed from ImageNet into YFCC, there can be two possibilities.
> - Possibility 1: YFCC models do not have better single-ID effective robustness compared to ImageNet models. It implies that the single-ID evaluation is problematic when there are models with different training data, as suggested in our paper.
> - Possibility 2: YFCC models still have better single-ID effective robustness compared to ImageNet models. Then it cannot directly reject the single-ID evaluation.
> As shown in Figure 1, the first possibility actually happens.
>
> Regarding the ImageNet accuracy overlap, it is similar to that in [12] which also compared the fitting lines (with ImageNet as the ID test set only). There is still an overlap at the mid-accuracy regime.
>
> ## Is the work novel?
> The core insight of our work is that apparent differences in effective robustness can be driven by differences in training sets and the evaluation needs to be aware of training data.  We believe that this is a novel insight, missed by many previous works on robustness [30, 12, 29], and leads to new conclusions and understanding of how to measure and improve robustness.  Therefore, the novelty lies not in how we empirically verify the insight, but in the insight itself.
>
> ## Generalization to more than two ID test sets
> In principle, a straightforward generalization is possible -- by directly generalizing Section 5.2, one can take the intersection of classes from more than two datasets. To check whether the evaluation is good enough, one can check if the fitting quality is satisfactory. In practice, this can be costly, as training much more models is needed to achieve a convincing fitting quality, when the evaluation has more than two datasets. It is not perfect, and we have already mentioned it as a limitation in the paper (Line 149-151, Line 316-317). Nevertheless, by comparing two datasets at a time in this work, we can already obtain conclusions that the considered models trained on any of the considered datasets have similar effective robustness, and a 3-way evaluation is not quite needed here.

---

> > ### Comment · Reviewer_mbLQ · 2023-08-19
> > **Thanks for the author's feedback while the concerns remain.**
> >
> > Thanks for your response! After reading the rebuttal and other reviews, I want to keep my previous score and tend to reject this submission.
> >
> > (1) **Motivation is unclear**. If the main goal of this paper is to make the robustness comparison between CLIP-based models (larger models trained with the contrastive loss with a large, diverse dataset) and traditional ImageNet-trained models with supervised loss, increasing one ID test to two or multiple ID test set is not a valid solution because it did not touch the underlying challenge. One critical challenge to avoid a fair comparison is missed during the whole discussion, which is the **training loss (contrastive loss v.s. cross-entropy loss) difference between CLIP-based models and traditional supervised learning trained models**.
> >
> > Yes, the large models that are trained on multiple training data do cover multiple domains, while they do not need strong supervision (only image-language pairs). The less requirement of the label makes it hard to define in-domain or OOD test set: given the training process are not classification task (contrastive loss), why use classification task to evaluate the robustness performance? The value the community places on the generalization ability of CLIP-based models isn't solely due to their superior performance on various test sets. It's also because they offer a viable solution for leveraging larger, more readily accessible datasets of weaker quality to achieve impressive results on diverse test sets.
> > In this case, if the proposed method does not consider the different training loss and requirements but only changes from using one test set to two or more test sets, the problem remains, which may mislead the whole community to keep missing the challenge of comparison between  CLIP based models and traditional supervised, trained models.
> >
> > (2) **Unclear of OOD test set** If it is hard to find a real OOD test set for the training set of the models that you want to explore (CLIP and LAION), how do you make the following effective robustness definition? The OOD test set is clear for the models that Taori et al [1] focused on, but for this submission's solution, which only increases the test set number while ignoring the faithfulness of the term they defined, it is confusing. The subsequent analysis seems unsupported by theory. In their rebuttal, the authors acknowledge the complexity of the issue but fail to offer a solution or enhancement. This heightens my concern about the experimental results lacking theoretical backing and analysis."
> >
> > (3) **Method lacks generalizability, and the experiment can not support the claim** As mentioned in the concerns, it is hard to generalize to more ID test sets. Moreover, defining an ID test set for CLIP trained on LAION is hard. Intuitively, it should depend on the real downstream task you care about. The experiments can not well support the paper's claim. The author's response did not solve my concern.
> > I also agree with reivewer Kcyx's concern on the unclear definition of multi-ID ER. It is unclear how to choose the numbers of in-domain test sets given a model trained on one dataset. If we compare the model with different groups of models, will the result change accordingly?
> >
> > I will keep my rejection score because of the reasons above.
> >
> >
> >
> > References
> >
> > [1] Taori, R., Dave, A., Shankar, V., Carlini, N., Recht, B., and Schmidt, L. Measuring robustness to natural distribution shifts in image classification. Advances in Neural Information Processing Systems, 33:18583–18599, 2020.

---

> > > ### Author Response · Authors · 2023-08-21
> > > **We are following many settings from prior works**
> > >
> > > We want to emphasize again that regarding most of the concerns on our settings, we are actually following prior works and our setup is well supported by the prior works.
> > >
> > > **Effect of the contrastive training loss?**
> > > The prior work [12] has already investigated and concluded that the exceptional single-ID effective robustness of CLIP models is not due to a contrastive loss but the difference in the training data (see Section 7 of [12]). Therefore, given this prior work, we are no longer considering the effect of the training loss here but focus on the training data.
> > >
> > > **OOD test sets**
> > > We have already clarified that our choice of the OOD test sets follows many prior works ([29], Schuhmann et al., 2022, Gadre et al., 2023).
> > >
> > > Therefore, we hope the reviewer could consider the conclusions and setup from multiple existing works. **We believe it is not valid to reject our paper due to the settings that we adopt from multiple prior works.**
> > >
> > > **Regarding "Method lacks generalizability, and the experiment can not support the claim"**:
> > > - Generalizing to multiple ID test sets is good to have but relatively minor, as we have already demonstrated that even if we compare two datasets at a time, it is already sufficient for us to reach the conclusions that models trained on multiple various datasets (ImageNet, YFCC, LAION) all have similar effective robustness. Thus directly comparing more than two datasets is not strongly necessary here.
> > > - We are focusing on the classification task in this work, following prior works [30, 12, 29]. To define an ID test set for CLIP trained on LAION, we are also using a classification task. We have demonstrated that even if we only use the classification task for the ID test set, it is already sufficient for us to predict the OOD accuracy well. Thus our treatment for the ID test set for LAION is already good enough.
> > > - Regarding the "unclear definition of multi-ID ER", please refer to our responses to Reviewer Kcyx. For "numbers of in-domain test sets", we propose to make the number the same as the number of training datasets involved. We have demonstrated that when we compare models trained on two different datasets each time, there are two ID sets (if all the models are trained on only one dataset, then the single-ID evaluation with only one ID test set is enough).
> > > - We are not sure what the reviewer specifically means by "if we compare the model with different groups of models, will the result change accordingly", but we have already demonstrated that the CLIP models or standard classification models trained on CIFAR-10, ImageNet, YFCC, LAION all have similar effective robustness, by comparing models trained on two datasets at each time. These models are commonly studied by prior works [30, 26, 12, 29].

---

### Decision · Program_Chairs · 2023-09-21

**Decision:**

Accept (poster)

**Comment:**

This paper aims to provide a more accurate comparison of the effective robustness (ER) of models trained on different datasets. Overall, the reviewers agree on the importance of the studied problem and find this paper interesting to read. Meanwhile, several major concerns are raised, including 1) additional ablations studies are needed, and 2) some claims/statements are confusing and probably incorrect.

The rebuttal was constructive --- with several rounds of discussions between authors and reviewers, several issues were identified and fixed accordingly, particularly regarding when the proposed multi-ID evaluation is applicable and how it could supplement existing single-ID evaluation for a better understanding of ER. All reviewers reached a unanimous decision to recommend the paper for acceptance.

For the paper to meet the rigorous publication standards of NeurIPS, the authors must undertake a significant revision of this paper, including rectifying flawed statements and incorporating the clarifications and ablation studies discussed during the rebuttal phase.